Letter

# The integrated stress response pathway controls cytokine production in tissue-resident memory CD4+ T cells

Nariaki Asada [1], Pauline Ginsberg[1], Hans-Joachim Paust[1], Ning Song[1], Jan-Hendrik Riedel[1], Jan-Eric Turner [1,2], Anett Peters[1], Anna Kaffke[1], Jonas Engesser[1], Huiying Wang[1], Yu Zhao [1,3], Robin Khatri [1,3], Philipp Gild[4], Roland Dahlem[4], Björn-Philipp Diercks [5], Sarada Das [6], Zoya Ignatova [6], Tobias B. Huber [1,2,7], Immo Prinz[2,8], Nicola Gagliani [2,9,10], Hans-Willi Mittrücker [2,11,12], Christian F. Krebs [1,2,7,12] & Ulf Panzer [1,2,7,12] ✉

Tissue-resident memory T ($T_{RM}$) cells are a specialized T cell population that reside in tissues and provide a rapid protective response upon activation. Here, we showed that human and mouse CD4+ $T_{RM}$ cells existed in a poised state and stored messenger RNAs encoding proinflammatory cytokines without protein production. At steady state, cytokine mRNA translation in $T_{RM}$ cells was suppressed by the integrated stress response (ISR) pathway. Upon activation, the central ISR regulator, eIF2α, was dephosphorylated and stored cytokine mRNA was translated for immediate cytokine production. Genetic or pharmacological activation of the ISR–eIF2α pathway reduced cytokine production and ameliorated autoimmune kidney disease in mice. Consistent with these results, the ISR pathway in CD4+ $T_{RM}$ cells was downregulated in patients with immune-mediated diseases of the kidney and the intestine compared to healthy controls. Our results indicated that stored cytokine mRNA and translational regulation in CD4+ $T_{RM}$ cells facilitate rapid cytokine production during local immune response.

T cell memory is a hallmark of the adaptive immune system that results from the antigen-specific activation, expansion and maintenance of T cells. Antigen-experienced memory T ($T_M$) cells elicit a faster and more potent immune response upon re-exposure to the same antigen and provide superior protection against reinfection than naive T ($T_N$) cells[1]. Based on their specific homing capacity and effector functions, circulating $T_M$ cells can be divided into central memory T ($T_{CM}$) and effector memory ($T_{EM}$) cells[1]. In contrast, tissue-resident memory T ($T_{RM}$) cells do not recirculate and exhibit long-term persistence at sites of previous infections[2–6]. $T_{RM}$ cells provide long-term immunity against local reinfections[2–6], but they also contribute to immune-mediated inflammatory diseases of the gut[7], central nervous system[8], skin[9,10] and kidney[11–13]. The local production of cytokines by CD4+ $T_{RM}$ cells

orchestrates protective tissue immunity against pathogens, but can also promote hyperinflammation and tissue damage in immune-mediated diseases[14–19].

Under homeostatic conditions, $T_M$ cells in nonlymphoid tissues express mRNA encoding effector molecules such as IFNγ and TNF for rapid cytokine production through translation of the already-transcribed mRNA[20–23]. Yet, whether different $T_M$ cell subsets differentially store cytokine mRNA remains incompletely explored. To address this issue, we combined single-cell RNA sequencing (scRNA-seq) and cellular indexing of transcriptomes and epitopes by sequencing (CITE-seq) to analyze CD3+CD4+ and CD3+CD8+ T cells from renal tissue and matched blood of eight healthy donors (cohort 1, median age 60.5 years (interquartile range (IQR) 55–72), 62.5% male)[13]. mRNA

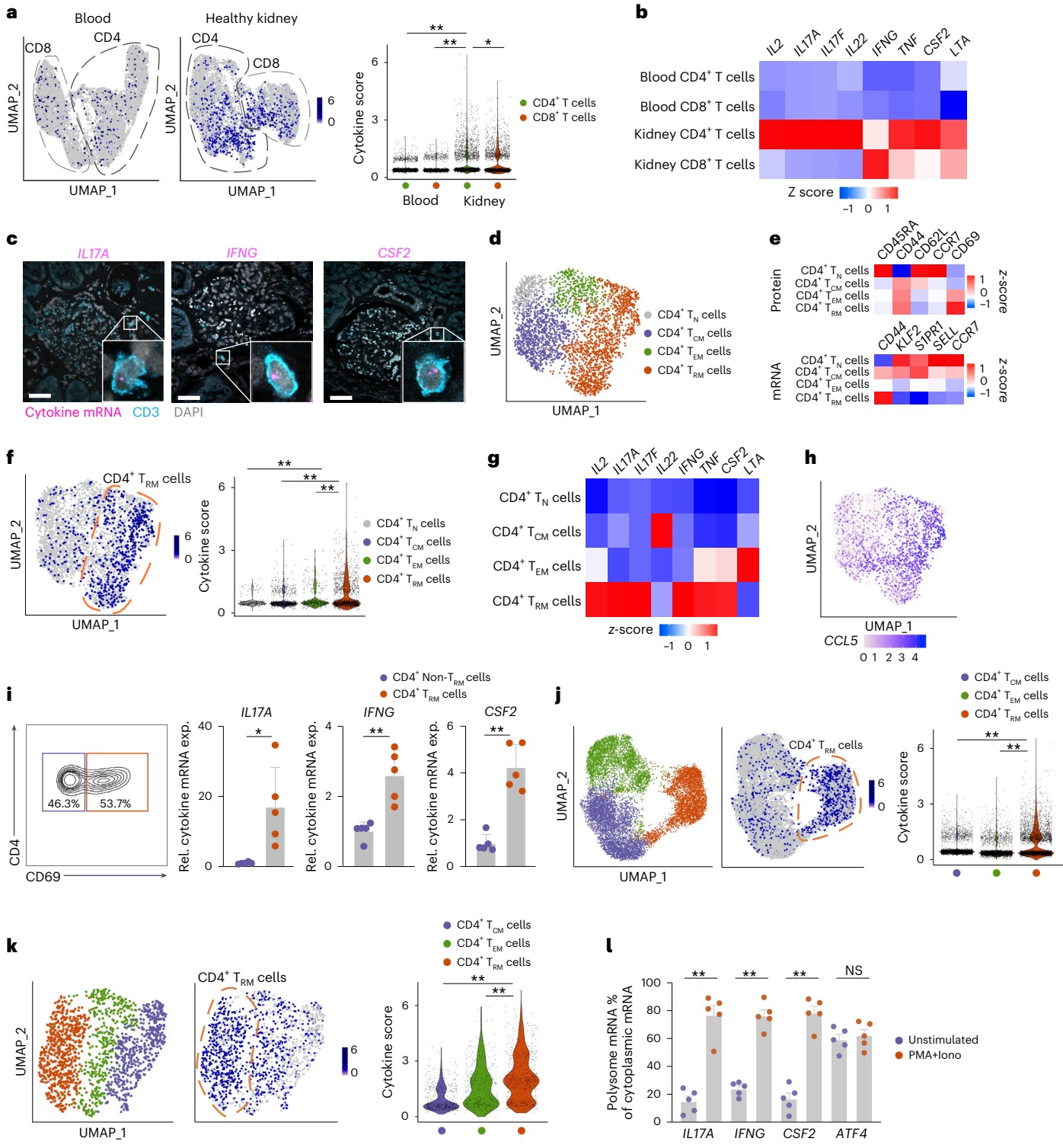

for type 1 (*IFNG*, *TNF*, *LTA* and *CSF2*) and type 3 (*IL17A*, *IL17F* and *IL22*) cytokines was detected in CD4⁺ and CD8⁺ T cells in the kidney, but was expressed at much lower levels in CD4⁺ and CD8⁺ T cells from the blood (Fig. 1a,b and Extended Data Fig. 1a,b). RNA fluorescence in situ hybridization (FISH) analysis detected CD3⁺ T cells expressing *IL17A*, *IFNG* and *CSF2* mRNA in healthy kidney tissue (Fig. 1c). Expression of *IL17A* and *IL17F* mRNA, and to a lesser degree of *CSF2* and *TNF* mRNA, was higher in CD4⁺ T cells compared to CD8⁺ T cells in the kidney (Fig. 1b).

Unsupervised clustering of human kidney CD4⁺ T cells identified CD45RA⁺ T_N, *CCR7⁺S1PR1⁺*CD44⁺ T_CM, *CCR7⁻S1PR1⁺*CD44⁺ T_EM and

*S1PR1⁻*CD44⁺CD69⁺ T_RM cells (Fig. 1d,e and Extended Data Fig. 1c). CD4⁺ T_RM cells were the predominant cellular source of proinflammatory cytokine mRNA (for example, *IL17A*, *IFNG* and *CSF2*) and chemokine mRNA (for example, *CCL4*, *CCL5* and *CCL20*) (Fig. 1f–h and Extended Data Fig. 1c,d). Quantitative PCR with reverse transcription (RT–qPCR) analysis indicated that *IL17A*, *IFNG* and *CSF2* mRNA expression was significantly higher in CD69⁺CD4⁺ T_RM cells compared to CD69⁻CD4⁺ non-T_RM cells isolated from the kidney of five healthy donors (cohort 2, median age 59 years (IQR 55–64), 60% male) (Fig. 1i). To explore whether CD4⁺ T_RM cells from other tissues expressed cytokine mRNA,

**Fig. 1 | CD4⁺ T_RM cells express proinflammatory cytokine mRNA without translation under homeostatic conditions. a**, Uniform Manifold Approximation and Projection (UMAP) plots showing cytokine scores based on the expression of *IL2*, *IL17A*, *IL17F*, *IL22*, *IFNG*, *TNF*, *CSF2* and *LTA* in CD4⁺ T cells and CD8⁺ T cells sorted from the blood and healthy kidney tissue of eight healthy donors (cohort 1). **b**, Heatmap showing the expression of the cytokine score genes in CD4⁺ T cells and CD8⁺ T cells from the blood and healthy kidney of donors in cohort 1. **c**, Representative images of mRNA FISH for *IL17A*, *IFNG* and *CSF2* combined with immunofluorescence staining for CD3 in healthy kidney tissue from human donors in cohort 1. Scale bars, 50 μm. The experiment was repeated three times. **d**, UMAP plot of CD4⁺ T cells isolated from human healthy kidney tissue (cohort 1), showing the distribution of *S1PR1*⁻CD44⁺CD69⁺ T_RM cells, *CCR7*⁻*S1PR1*⁺CD44⁺ T_EM cells, *CCR7*⁺*S1PR1*⁺CD44⁺ T_CM cells and CD45RA⁺ T_N cells. **e**, Heatmaps showing the protein expression of CD45RA, CD44, CD62L, CCR7 and CD69 (top) and the mRNA expression of *CD44*, *KLF2*, *S1PR1*, *SELL* and *CCR7* (bottom) in CD4⁺ T_RM, T_EM, T_CM and T_N cells (as in **d**). **f**, UMAP plot and violin plot showing cytokine scores in CD4⁺ T_RM, T_EM, T_CM and T_N cells (as in **d**). **g**, Heatmap

showing the expression of the cytokine score genes in CD4⁺ T_RM, T_EM, T_CM and T_N cells (as in **d**). **h**, UMAP plot showing *CCL5* mRNA expression in CD4⁺ T_RM, T_EM, T_CM and T_N cells (as in **d**). **i**, Representative flow cytometry plot showing the gating strategy of CD4⁺CD69⁺ T_RM cells and CD4⁺CD69⁻ non-T_RM cells and RT–qPCR analysis showing the expression of *IL17A*, *IFNG* and *CSF2* mRNA in CD69⁺CD4⁺ T_RM cells and CD69⁻CD4⁺ non-T_RM cells sorted from the human healthy kidneys (*n* = 5, cohort 2). **j,k**, UMAP plots showing CD4⁺ T cells (left) and violin plots showing cytokine scores in *SELL*⁻*CCR7*⁻*KLF2*⁻ T_RM cells, *SELL*⁻*CCR7*⁻*KLF2*⁺ T_EM cells and *SELL*⁺*CCR7*⁺*S1PR1*⁺ T_CM cells (middle and right) in colon[24] (*n* = 4) (**j**) and lung[25] (*n* = 10) (**k**) (left) from healthy human donors. **l**, Percentage of polysome (translated) mRNA out of cytoplasmic RNA based on RT–qPCR quantification of *IL17A*, *IFNG*, *CSF2* and *ATF4* transcripts in CD4⁺ T_RM cells isolated from human healthy kidneys and left unstimulated or stimulated with PMA + Iono for 1 h (*n* = 5). Data are mean + s.e.m. Statistical analysis was performed using a one-way analysis of variance (ANOVA) with Tukey's multiple comparison test (**a,f,j**) or unpaired two-tailed *t*-test with Welch's correction (**i,l**); *$P < 0.05$, **$P < 0.01$. NS, not significant.

we analyzed public scRNA-seq datasets for healthy human colon[24] and lung[25]. *CD4*⁺*SELL*⁻*CCR7*⁻*KLF2*⁻ T_RM cells were detected in the healthy colon (Fig. 1j) and lung (Fig. 1k), and expressed higher levels of cytokine mRNA, such as *IFNG*, *TNF* and *CSF2*, compared to other T_M cell populations (Fig. 1j,k and Extended Data Fig. 2a–d). These data showed that CD4⁺ T_RM cells expressed proinflammatory cytokine mRNA in various human tissues under noninflammatory conditions.

Production of inflammatory cytokines might lead to inflammation and immunopathology. In the kidney tissue where CD4⁺ T_RM cells were isolated, we did not observe any signs of active inflammation, such as immune cell infiltration (Extended Data Fig. 3a). To investigate whether cytokine mRNA expression in CD4⁺ T_RM cells led to cytokine production, we isolated CD4⁺ T cells from the blood and kidney of five healthy donors (cohort 2). These cells were treated with brefeldin A, an inhibitor of vesicle transport between the endoplasmic reticulum and the Golgi apparatus, in the presence or absence of phorbol 12-myristate 13-acetate (PMA) + ionomycin (Iono), which mimic T cell receptor activation, for 6 h. Stimulated kidney CD4⁺ T cells, but not unstimulated ones, produced interleukin (IL)-17A, interferon (IFN)γ and granulocyte–macrophage colony-stimulating factor (GM-CSF) protein (Extended Data Fig. 3b). To investigate whether expression of proinflammatory cytokine mRNA in T cells gave rise to tissue responses under homeostatic conditions, we analyzed the European Renal cDNA Bank database[26], which encompasses kidney biopsies from healthy individuals (living donor kidney transplantation) and from patients with autoimmune kidney diseases, such as anti-neutrophil cytoplasmic antibody-associated glomerulonephritis (ANCA-GN), lupus nephritis and IgA nephropathy. We found high expression of genes induced by IL-17, IFNγ or GM-CSF signaling, such as *CXCL1* and *CXCL5* (IL-17-responsive genes), *CXCL9* and *CXCL10* (IFNγ-responsive genes) and *CCR2* and *CCL22* (GM-CSF-responsive genes) in the kidney

tissue from autoimmune kidney disease patients[27–29] (Extended Data Fig. 3c,d). In contrast, expression of these genes was significantly lower in healthy kidney tissue (Extended Data Fig. 3c,d), suggesting that the kidney does not exhibit a tissue response to cytokines under homeostatic conditions.

To investigate the translation status of the cytokine mRNA in resting CD4⁺ T_RM cells, CD69⁺CD4⁺ T_RM cells were isolated from healthy human kidney tissue. These cells were left unstimulated or stimulated with PMA + Iono for 1 h and mixed with 2 million NIH/3T3 cells to achieve sufficient cell numbers to identify the border between the translated and untranslated mRNA fractions, referred to as the monosome peak. The cells were lysed and subjected to sucrose gradient centrifugation to separate the translated (polysome) and untranslated mRNA fractions[30] for RT–qPCR analysis. We found that only around 20% of *IL17A*, *IFNG* and *CSF2* mRNA was detected in the translated mRNA fraction in unstimulated CD69⁺CD4⁺ T_RM cells, whereas approximately 80% was found in the translated mRNA fraction in CD69⁺CD4⁺ T_RM cells stimulated with PMA + Iono (Fig. 1l), indicating that proinflammatory cytokine mRNAs were not translated in CD69⁺CD4⁺ T_RM cells at steady state. *ATF4* mRNA, which is translated under compromised cap-dependent translation[31], was efficiently translated in unstimulated CD69⁺CD4⁺ T_RM cells (Fig. 1l), indicating that the translation of cytokine mRNAs was actively suppressed in CD4⁺ T_RM cells under homeostatic conditions.

Mice maintained under specific-pathogen-free conditions harbor few T_RM cells in nonbarrier organs such as the kidney. To perform an in-depth functional analysis of CD4⁺ T_RM cells, we induced kidney T_RM cells[13], particularly IL-17A-producing T_RM (T_RM17) cells, by systemically infecting *Il17a*^Cre*R26*^eYFP mice[32], in which IL-17A-producing cells are labeled with yellow fluorescent protein (YFP), with intravenously injected *Staphylococcus aureus*. Starting at day 7 post-infection, we treated the mice with antibiotics for 2 weeks to clear the infection.

**Fig. 2 | Murine T_RM cells show downregulated mRNA translation profile under homeostasis with immediate cytokine production ability upon stimulation. a**, Representative images of mRNA FISH of *Il17a*, *Ifng* and *Csf2* mRNA combined with immunofluorescence CD3 staining in the kidney of IL-17A^MI mice. Scale bars, 50 μm. The experiment was repeated three times. **b**, UMAP plots showing the distribution of *Sell*⁻*S1pr1*⁻*Klf2*⁻ T_RM17 cells, *Sell*⁻*S1pr1*⁺*Klf2*⁺ T_EM17 cells and *Sell*⁺*S1pr1*⁺*Klf2*⁺ T_CM17 cells in YFP⁺CD4⁺ T cells isolated from the kidneys of IL-17A^MI mice and analyzed by scRNA-seq (left) and violin plot showing cytokine scores calculated based on the expression of *Il2*, *Il17a*, *Il17f*, *Il22*, *Ifng*, *Tnf*, *Csf2* and *Lta* in the YFP⁺CD4⁺ T_RM17, T_EM17 and T_CM17 cell clusters (middle and right). **c**, Heatmap showing the expression of *Il2*, *Il17a*, *Il17f*, *Il22*, *Ifng*, *Tnf*, *Csf2* and *Lta* in YFP⁺CD4⁺ T_RM17, T_EM17 and T_CM17 cell clusters as in **b**. **d**, GO analysis displaying pathways downregulated in YFP⁺CD4⁺ T_RM17 cells compared to YFP⁺CD4⁺ T_EM17 cells as in **b**. **e**, Violin plot showing the ribosomal protein gene score calculated based on the expression of all RPGs in YFP⁺CD4⁺ T_RM17, T_EM17 and T_CM17 cell clusters as in

**b**. **f**, Volcano plot showing differential gene expression between YFP⁺CD4⁺ T_RM17 and YFP⁺CD4⁺ T_EM17 cells as in **b**. RPGs downregulated in T_RM17 cells are shown in orange. **g**, Time course of IL-17A production in YFP⁺CD4⁺CD44⁺CD69⁺ T_RM17 cells, YFP⁺CD4⁺CD44⁺CD69⁻CD62L⁻ T_EM17 cells, YFP⁺CD4⁺CD44⁺CD62L⁺ T_CM17 cells and YFP⁻CD4⁺CD44⁻ T_N cells isolated from the kidneys of IL-17A^MI mice and stimulated with PMA + Iono for 30 min, 1 h and 3 h (*n* = 6). **h**, Representative flow cytometry plots showing IL-17A production in YFP⁺CD4⁺CD44⁺CD69⁺ T_RM17 cells and YFP⁺CD4⁺CD44⁺CD69⁻CD62L⁻ T_EM17 cells isolated from the kidneys of IL-17A^MI mice stimulated with PMA + Iono in the presence or absence of 20 μg ml⁻¹ actinomycin D for 4 h (*n* = 3). **i**, Representative flow cytometry plots showing IL-17A production in CD4⁺CD44⁺CD69⁺ T_RM cells and CD4⁺CD44⁺CD69⁻CD62L⁻ T_EM cells isolated from the intestine of wild-type C57BL/6 mice stimulated with PMA + Iono in the presence or absence of 20 μg ml⁻¹ actinomycin D for 4 h (*n* = 4). Data are mean + s.e.m. Statistical analysis was performed using one-way ANOVA with Tukey's multiple comparison test; **$P < 0.01$.

Mice were analyzed 2 months after the end of the antibiotics treatment[13,33]. We refer to these mice as memory-induced IL-17A reporter mice (hereafter IL-17A[MI] mice). Immunofluorescence staining combined with mRNA FISH detected numerous CD3[+] T cells in the kidney of IL-17A[MI] mice, with some cells expressing *Il17a*, *Ifng* or *Csf2* (Fig. 2a). scRNA-seq analysis on YFP[+]CD4[+] T cells sorted from the kidneys of IL-17A[MI] identified *Sell[+]S1pr1[+]Klf2[+]* $T_{CM}$ cell (hereafter CD4[+] TCM17 cell), *Sell[+]S1pr1[+]Klf2[+]* $T_{EM}$ cell (CD4[+] $T_{EM}$17 cell) and *Sell[-]S1pr1[-]Klf2[-]* $T_{RM}$ cell (CD4[+] $T_{RM}$17 cell) subsets (Fig. 2b and Extended Data Fig. 4a).

CD4[+] $T_{RM}$17 cells highly expressed mRNA for *Il2*, *Il17a*, *Il17f*, *Il22*, *Ifng*, *Tnf*, *Csf2* and *Lta*, while CD4[+] $T_{CM}$17 cells expressed only *Csf2* and CD4[+] $T_{EM}$17 cell expressed only *Il22* and *Lta* (Fig. 2b,c and Extended Data Fig. 4b). RT−qPCR confirmed the higher expression of *Il17a* mRNA in kidney YFP[+]CD4[+]CD44[+]CD69[+]CD62L[-] $T_{RM}$17 cells compared to YFP[+]CD4[+]CD44[+]CD69[-]CD62L[-] $T_{EM}$17 cells, YFP[+]CD4[+]CD44[+]CD69[-]CD62L[+] $T_{CM}$17 cells, YFP[-]CD4[+]CD44[+]CD69[+]CD62L[-] $T_{RM}$ cells and CD4[+]CD44[-] $T_{N}$ cells (Extended Data Fig. 4c). Gene Ontology (GO) pathway analysis indicated that pathways associated with cytoplasmic

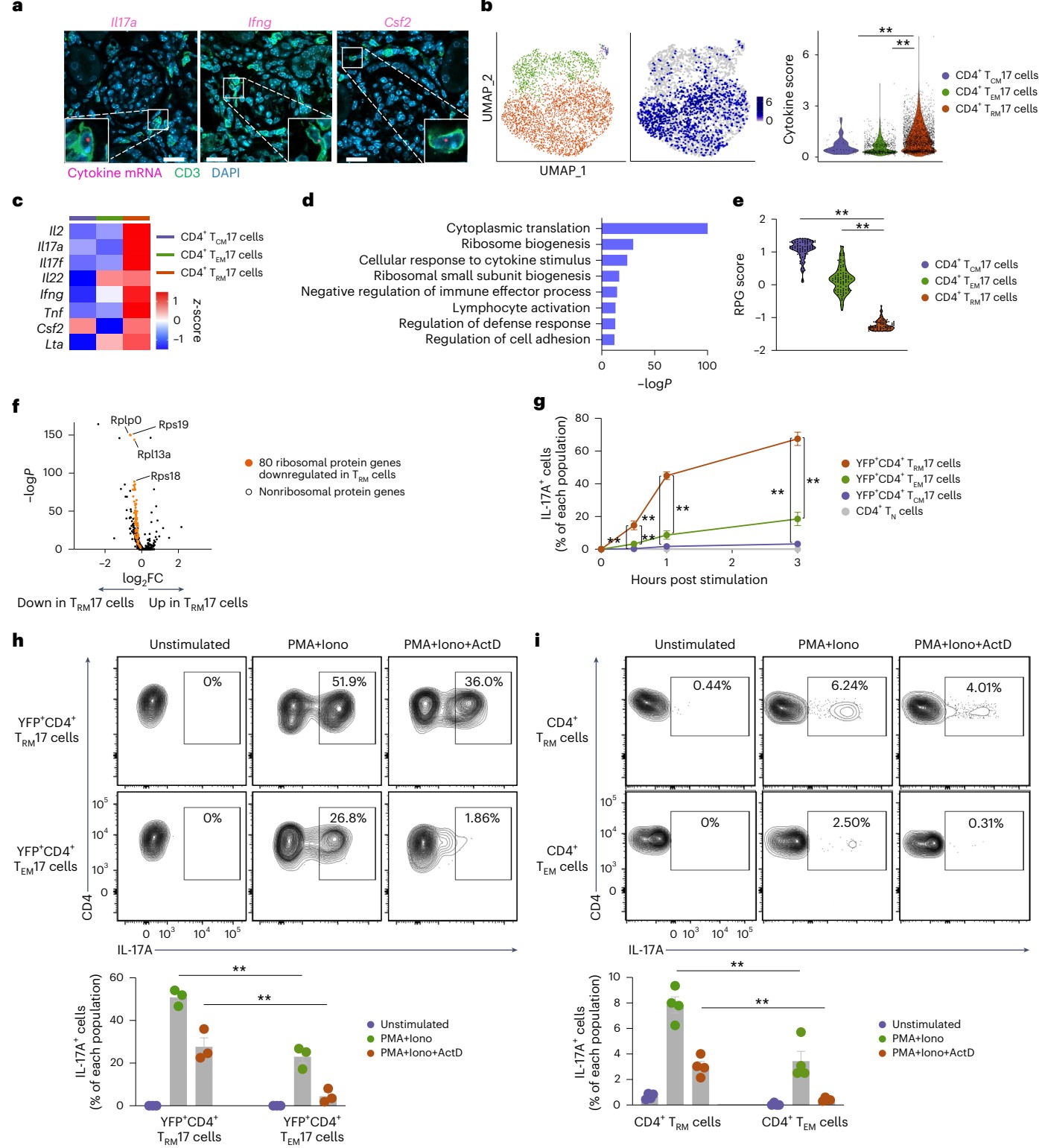

translation and ribosome biogenesis were significantly downregulated in CD4$^+$ T$_{RM}$17 cells compared to CD4$^+$ T$_{EM}$17 cells (Fig. 2d). In the scRNA-seq data of YFP$^+$CD4$^+$ T cells, we observed the coordinated downregulation of ribosomal protein genes (RPGs)[34], such as *Rpsa*, *Rps3* and *Rpl3* in CD4$^+$ T$_{RM}$17 cells compared to CD4$^+$ T$_{EM}$17 cells and CD4$^+$ T$_{CM}$17 cells (Fig. 2e,f and Extended Data Fig. 4d), indicating the suppression of global mRNA translation in CD4$^+$ T$_{RM}$17 cells. Moreover, the protein–protein interaction enrichment analysis[35] indicated suppression of the ribosomal scanning and start codon recognition processes, which are crucial steps in translation initiation, in CD4$^+$ T$_{RM}$17 cells compared to CD4$^+$ T$_{EM}$17 cells and CD4$^+$ T$_{CM}$17 cells (Extended Data Fig. 4e). These findings indicated that CD4$^+$ T$_{RM}$17 cells from the kidney of IL-17A$^{MI}$ mice express cytokine mRNA while suppressing mRNA translation under homeostatic conditions.

Rapid production of cytokines by T$_M$ cells is crucial for effective host defense. To investigate the kinetics of cytokine production, we stimulated YFP$^+$CD4$^+$CD44$^+$CD69$^+$ T$_{RM}$17 cells, YFP$^+$CD4$^+$CD44$^+$CD69$^-$CD62L$^-$ T$_{EM}$17 cells, YFP$^+$CD4$^+$CD44$^+$CD62L$^+$ T$_{CM}$17 cells and YFP$^-$CD4$^+$CD44$^-$ T$_N$ cells from the kidneys of IL-17A$^{MI}$ mice with PMA + Iono. CD4$^+$ T$_{RM}$17 cells produced higher levels of IL-17A, IFNγ and GM-CSF at 30 min compared to CD4$^+$ T$_{CM}$17 cells and CD4$^+$ T$_{EM}$17 cells (Fig. 2g and Extended Data Fig. 4f). To investigate whether the rapid production of IL-17A in CD4$^+$ T$_{RM}$17 cells was due to the translation of pre-transcribed mRNA, we stimulated CD4$^+$ T$_{RM}$17 cells and CD4$^+$ T$_{EM}$17 cells from the kidneys of IL-17A$^{MI}$ mice with PMA + Iono in the presence or absence of actinomycin D (ActD), which blocks mRNA transcription. CD4$^+$ T$_{RM}$17 cells and, to a lesser extent, CD4$^+$ T$_{EM}$17 cells produced IL-17A upon stimulation with PMA + Iono in the absence of ActD (Fig. 2h). When stimulated with PMA + Iono + ActD, CD4$^+$ T$_{RM}$17 cells (but not CD4$^+$ T$_{EM}$17 cells) still produced IL-17A protein, although at lower levels than those stimulated with PMA + Iono alone (Fig. 2h). Similar results were observed in CD4$^+$CD44$^+$CD69$^+$ T$_{RM}$ cells and CD4$^+$CD44$^+$CD69$^-$CD62L$^-$ T$_{EM}$ cells from the small intestine of healthy wild-type mice (Fig. 2i). These findings indicated that transcribed *Il17a* mRNA pools stored in CD4$^+$ T$_{RM}$ cells were sufficient for immediate cytokine protein production upon stimulation.

To understand the molecular pathways that regulated mRNA translation in human CD4$^+$ T$_{RM}$ cells, we analyzed the scRNA-seq dataset of kidney T cells (cohort 1) and published datasets of colon[24] and lung[25] T cells. GO term pathway analysis and ribosomal protein gene expression analysis revealed the downregulation of translation and ribosome biogenesis pathways in healthy human CD4$^+$ T$_{RM}$ cells isolated from the kidney (Fig. 3a,b and Extended Data Fig. 5a), colon and lung (Extended Data Fig. 5b–e) compared to other healthy human CD4$^+$ T$_M$ cell subsets. In contrast, Reactome pathway analysis found upregulation of pathways linked to cellular responses to stress in human kidney CD4$^+$ T$_{RM}$ cells (Fig. 3c). Cellular stress responses are associated with the integrated stress response (ISR) pathway, a biological process activated by various stimuli, such as hypoxia and unfolded protein response[31,36]. The ISR phosphorylates eIF2α to decrease cap-dependent translation initiation and downregulate global mRNA translation[31]. Similarly, analysis of the scRNA-seq dataset in the IL-17A$^{MI}$ mice also revealed the downregulation of the start codon recognition and translation initiation process in CD4$^+$ T$_{RM}$17 cells compared to other CD4$^+$ T$_M$17 cells (Extended Data Fig. 4e). Consistent with these findings, scRNA-seq analysis of human CD4$^+$ T cells from the healthy kidney, colon and lung showed that human CD4$^+$ T$_{RM}$ cells expressed high levels of ISR-associated genes, such as *PPP1R15A*, *ATF3*, *ATF4* and *ATF5* (refs. 31,37–41) (Fig. 3d and Extended Data Fig. 5f–i). eIF2α was phosphorylated in unstimulated human kidney CD4$^+$CD69$^+$ T$_{RM}$ cells, but was dephosphorylated after stimulation of these cells with PMA + Iono for 3 h or CD3$^+$ CD28 Ab for 2 days (Fig. 3f,g and Extended Data Fig. 5j,k). In a puromycin-uptake assay, unstimulated human kidney CD4$^+$CD69$^+$ T$_{RM}$ cells exhibited lower levels of global protein synthesis compared to CD4$^+$CD69$^+$ T$_{RM}$ cells stimulated with PMA + Iono for 2 h (Extended Data Fig. 5l).

Similar to human CD4$^+$ T$_{RM}$ cells, unstimulated kidney CD4$^+$CD69$^+$ T$_{RM}$ cells from memory-induced wild-type C57BL/6 mice (hereafter WT$^{MI}$ mice) showed high p-eIF2α levels during homeostasis, while following stimulation with PMA + Iono for 3 h or CD3 + CD28 antibody for 2 days, these cells exhibited eIF2α dephosphorylation (Fig. 4a,b and Extended Data Fig. 6a). In CD4$^+$CD69$^+$ T$_{RM}$ cells stimulated with PMA + Iono, global protein synthesis significantly increased compared to unstimulated controls (Extended Data Fig. 6b). In time kinetic experiments using kidney CD4$^+$CD69$^+$ T$_{RM}$ cells from WT$^{MI}$ mice, eIF2α dephosphorylation was observed 15 min after PMA + Iono stimulation (Fig. 4c). Cytokine production was detected 30 min after stimulation (Fig. 4c). p-eIF2α levels were significantly lower in IL-17A- or IFNγ-producing CD4$^+$CD69$^+$ T$_{RM}$ cells stimulated with PMA + Iono compared to unstimulated controls (Extended Data Fig. 6c). Additionally, mouse CD4$^+$CD69$^+$ T$_{RM}$ cells had higher p-eIF2α levels than CD4$^+$CD44$^+$CD69$^-$CD62L$^+$ T$_{CM}$ cells, CD4$^+$CD44$^+$CD69$^-$CD62L$^-$ T$_{EM}$ cells and CD4$^+$CD44$^-$ T$_N$ cells (Extended Data Fig. 6d). Following ISR activation, untranslated mRNAs are sequestered into stress granules, which are membrane-less cytoplasmic organelles[30]. *Il17a* mRNA FISH analysis combined with immunocytochemistry for the stress granule marker Tia1 revealed increased colocalization of cytoplasmic *Il17a* mRNA and Tia1 in unstimulated YFP$^+$CD4$^+$CD69$^+$ T$_{RM}$ cells from IL-17A$^{MI}$ mice compared to YFP$^+$CD4$^+$CD69$^+$ T$_{RM}$ cells stimulated with PMA + Iono for 3 h (Fig. 4d), indicating the release of *Il17a* mRNA from the stress granules in stimulated T$_{RM}$ cells.

To investigate the role of the ISR pathway in proinflammatory cytokine production in CD4$^+$CD69$^+$ T$_{RM}$ cells from WT$^{MI}$ mice, we used sodium arsenite, which activates the ISR pathway by oxidative cues[30]. Sodium arsenite treatment significantly increased eIF2α phosphorylation (Fig. 4e) and suppressed the production of IL-17A, IFNγ and GM-CSF in CD4$^+$CD69$^+$ T$_{RM}$ cells from WT$^{MI}$ mice stimulated with PMA + Iono (Fig. 4f and Extended Data Fig. 6e,f). Sodium arsenite treatment did not affect cell viability or intracellular Ca$^{2+}$ concentration increase upon T cell stimulation (Extended Data Fig. 6g,h), indicating no signs of toxicity or inhibition of activation. These observations indicated that the ISR controlled the translation of cytokine mRNA.

We next tested whether blocking eIF2α dephosphorylation suppresses cytokine mRNA translation in activated T$_{RM}$ cells. eIF2α is dephosphorylated by the phosphatase complexes PPP1R15A-PP1 and PPP1R15B-PP1 (ref. 31). The pan-eIF2α phosphatase inhibitor Sal003, which blocks both PPP1R15A and PPP1R15B, significantly suppressed eIF2α dephosphorylation and the production of IL-17A, IFNγ and GM-CSF in CD4$^+$CD69$^+$ T$_{RM}$ cells from WT$^{MI}$ mice stimulated with PMA + Iono (Fig. 4e and Extended Data Fig. 6e, f). Sephin1, a selective inhibitor of PPP1R15A, did not suppress eIF2α dephosphorylation or the production of IL-17A, IFNγ and GM-CSF in CD4$^+$CD69$^+$ T$_{RM}$ cells (Fig. 4e and Extended Data Fig. 6e,f). In contrast, raphin1, a selective inhibitor of PPP1R15B, significantly reduced eIF2α dephosphorylation and suppressed the production of those cytokines (Fig. 4e and Extended Data Fig. 6e,f), indicating eIF2α dephosphorylation was predominantly driven by PPP1R15B. None of these compounds suppressed intracellular Ca$^{2+}$ concentration increase or shedding of surface CD62L upon T cell activation (Extended Data Fig. 6h,i), indicating that T cell activation was not affected. In addition, CD4$^+$CD69$^+$ T$_{RM}$ cells from WT$^{MI}$ mice treated with ISRIB, which binds to and activates eIF2B to facilitate mRNA translation during the ISR, showed a moderate increase in IL-17A and IFNγ production compared to untreated WT$^{MI}$ CD4$^+$CD69$^+$ T$_{RM}$ cells in the absence of TCR stimulation (Extended Data Fig. 6j). We also analyzed CD8$^+$CD69$^+$ T$_{RM}$ cells isolated from the intestine of wild-type C57BL/6 mice. eIF2α was phosphorylated in unstimulated CD8$^+$CD69$^+$ T$_{RM}$ cells, while after stimulation with PMA + Iono for 3 h, CD8$^+$CD69$^+$ T$_{RM}$ cells dephosphorylated eIF2α and produced of IFNγ (Extended Data Fig. 6k). Sodium arsenite treatment increased p-eIF2α levels and inhibited IFNγ production in these cells (Extended

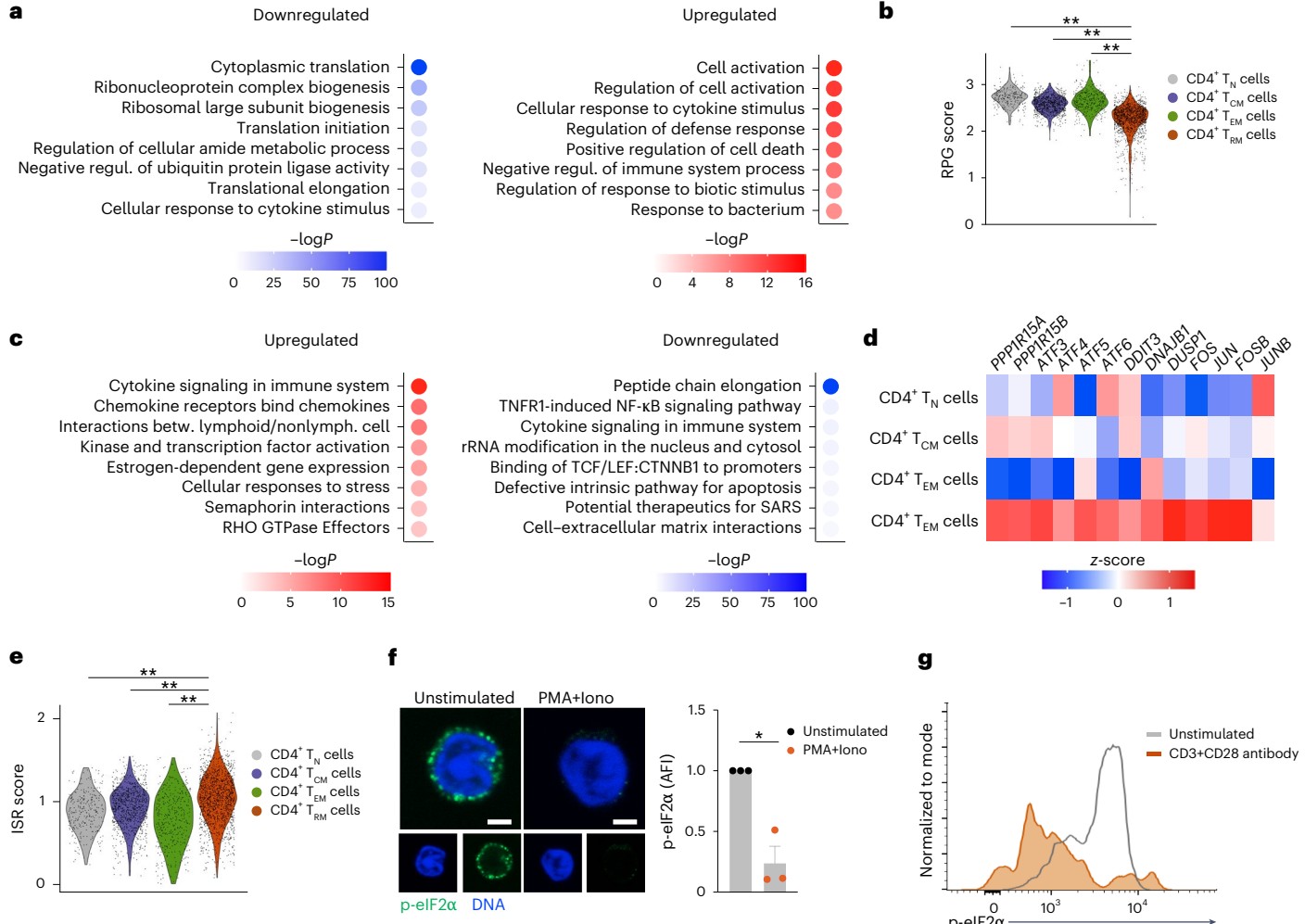

**Fig. 3 | The integrated stress response pathway is activated in human T_RM cells.** **a**, GO analysis showing pathways downregulated (left) and upregulated (right) in *CD4⁺SELL⁻S1PR1⁻*CD69⁺ T_RM cells compared to CD4⁺CD69⁻ non-T_RM cells isolated from human healthy kidneys (scRNA-seq data of cohort 1 as in Fig. 1d). **b**, Violin plot showing the ribosomal protein gene score calculated based on the expression of all RPGs in CD4⁺ T_RM, T_EM, T_CM and T_N cell clusters (as in **a**). **c**, Reactome analysis displaying pathways upregulated (left) and downregulated (right) in *CD4⁺SELL⁻S1PR1⁻*CD69⁺ T_RM cells compared to CD4⁺CD69⁻ non-T_RM cells in human healthy kidneys (as in **a**). **d**, Heatmap showing the expression of the ISR-associated genes *PPP1R15A, PPP1R15B, ATF3, ATF4, ATF5, ATF6, DDIT3, DNAJB1, DUSP1, FOS, JUN, FOSB* and *JUNB* in CD4⁺ T_RM, T_EM, T_CM and T_N cell clusters (as in **a**). **e**, Violin plot showing the ISR-associated gene scores in CD4⁺ T_RM, T_EM,

T_CM and T_N cell clusters (as in **a**). **f**, Representative immunocytochemistry images of CD4⁺CD69⁺ T_RM cells sorted from healthy human kidney and stained with antibodies specific for phosphorylated eIF2α (p-eIF2α) and 4,6-diamidino-2-phenylindole (DAPI) and quantification of p-eIF2α expression in CD4⁺CD69⁺ T_RM cells sorted from healthy human kidney and stimulated or not with PMA + Iono for 3 h (*n* = 3). Scale bars, 2 μm. **g**, Representative histogram showing p-eIF2α levels in CD4⁺CD69⁺ T_RM cells with or without CD3 + CD28 antibody stimulation for 2 days (representative of two experiments). Data are mean + s.e.m. Statistical analysis was performed using one-way ANOVA with Tukey's multiple comparison test (**b**,**e**) or unpaired two-tailed *t*-test with Welch's correction (**f**); *P < 0.05, **P < 0.01.

Data Fig. 6k). These findings indicated that the ISR pathway might regulate cytokine mRNA translation in CD8⁺ T_RM cells.

To further test whether eIF2α phosphorylation was required for the suppression of cytokine production, we transduced CD4⁺CD69⁺ T_RM cells from WT^MI mice with a constitutively active (eIF2α-S51A) or a constitutively inactive (eIF2α-S51D, a phosphomimetic variant) form of eIF2α[42]. Production of IL-17A was significantly suppressed in eIF2α-S51D-transduced T cells compared to eIF2α-S51A-transduced CD4⁺CD69⁺ T_RM cells (Fig. 4f), indicating that eIF2α phosphorylation was sufficient to suppress IL-17A production in mouse CD4⁺CD69⁺ T_RM cells. Next, we transferred either eIF2α-S51D-transduced or eIF2α-S51A-transduced CD4⁺CD69⁺ T_RM cells into T cell-deficient *Rag1* KO mice. One day after transfer, the mice were intraperitoneally injected with nephrotoxic serum to induce autoimmune glomerulonephritis (crescentic GN; cGN)[13]. At day 10 post-cGN induction, mice that received eIF2α-S51D-transduced CD4⁺ T cells showed a milder

renal disease phenotype, as assessed by glomerular crescent scoring and albuminuria measurement, compared to mice that received eIF2α-S51A-transduced CD4⁺ T cells (Fig. 4f). These findings indicated that targeting the ISR pathway in CD4⁺ T cells ameliorated tissue inflammation and cGN progression.

To further investigate the therapeutic effects of targeting eIF2α dephosphorylation in autoimmune disease, we induced cGN in WT^MI mice (hereafter cGN^MI mice) and treated them with raphin1 (ref. 43) via twice-daily oral gavage. At day 10 of cGN, the levels of p-eIF2α in kidney CD4⁺CD69⁺ T_RM cells were higher in raphin1-treated compared to vehicle-treated cGN^MI mice (Extended Data Fig. 7a). Raphin1-treated cGN^MI mice exhibited reduced glomerular crescent formation and albuminuria, indicating improved kidney function compared to vehicle-treated cGN^MI mice (Fig. 4g,h). Raphin1 treatment did not improve kidney function in nephritic wild-type mice in which CD4⁺CD69⁺ T_RM cells were not induced by infection (Extended

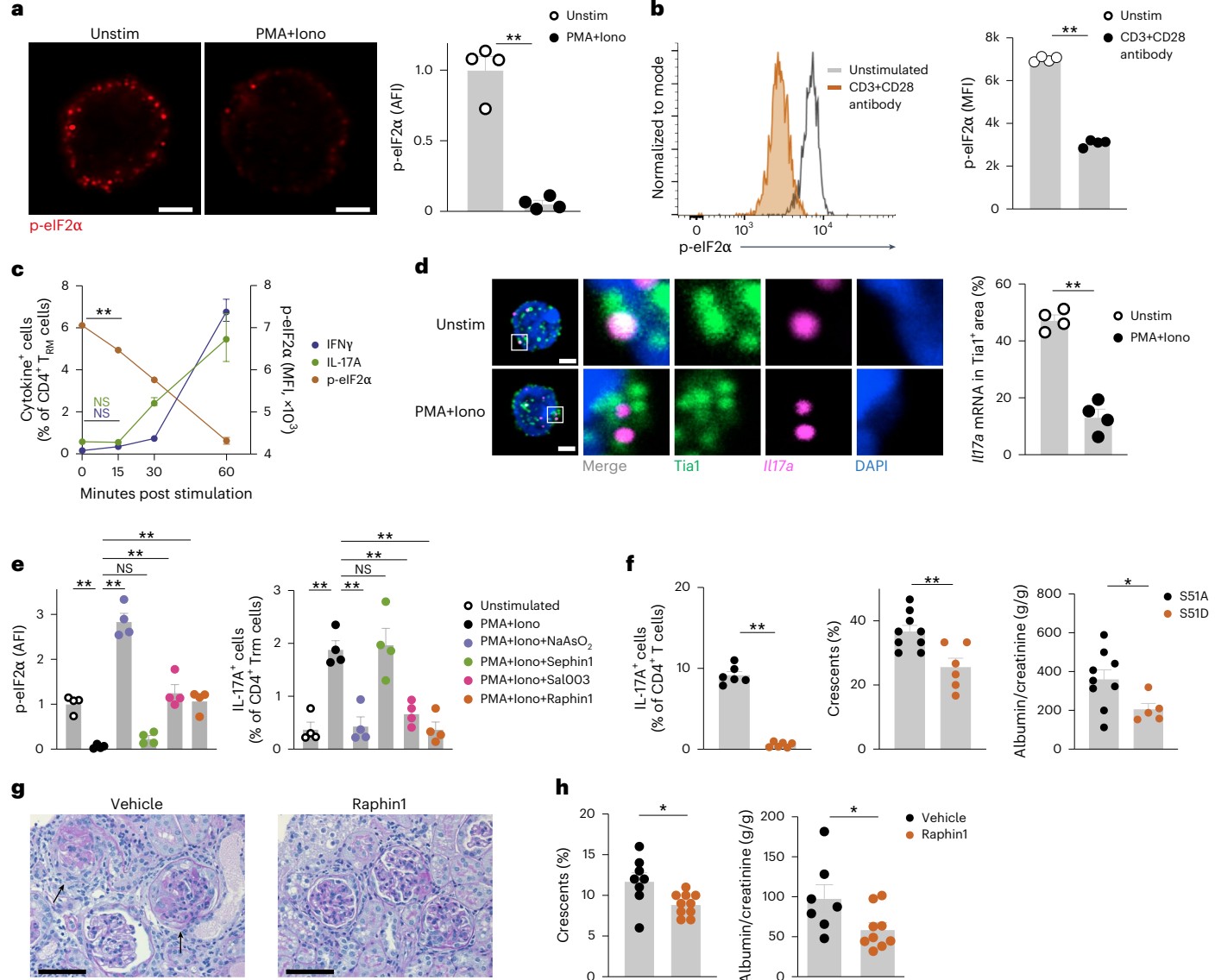

**Fig. 4 | The integrated stress response pathway regulates cytokine production. a**, Representative immunocytochemistry of CD4⁺CD69⁺ T$_{RM}$ cells isolated from the kidneys of C57BL/6 WT$^{MI}$ mice and stained with antibodies specific for phosphorylated eIF2α (p-eIF2α) and DAPI, and quantification of p-eIF2α expression in CD4⁺CD69⁺ T$_{RM}$ cells isolated from the kidneys of C57BL/6 WT$^{MI}$ mice stimulated or not with PMA + Iono for 3 h ($n = 4$). Scale bars, 2 μm. **b**, Representative histogram and quantification of p-eIF2α in CD4⁺CD69⁺ T$_{RM}$ cells isolated from the kidneys of WT$^{MI}$ mice and stimulated or not with CD3 + CD28 antibody for 2 days ($n = 4$). **c**, Time course plot showing the frequency of IFNγ⁺ and IL-17A⁺ in CD4⁺CD69⁺ T$_{RM}$ cells and p-eIF2α expression in CD4⁺CD69⁺ T$_{RM}$ cells isolated from the kidneys of WT$^{MI}$ mice and stimulated with PMA + Iono for 0, 15, 30 and 60 min ($n = 4$). **d**, Representative mRNA FISH for *Il17a* mRNA combined with immunocytochemistry Tia1 staining in YFP⁺CD4⁺CD69⁺ T$_{RM}$17 cells sorted from the kidneys of IL-17A$^{MI}$ mice and stimulated or not with PMA + Iono for 3 h (left), and quantification of *Il17a* mRNA and Tia1 colocalization in YFP⁺CD4⁺CD69⁺ T$_{RM}$17 cells sorted from the kidneys of IL-17A$^{MI}$ mice and stimulated or not with

PMA + Iono for 3 h (right). Scale bars, 2 μm. Quantification of four experiments is shown. **e**, Expression of p-eIF2α and frequency of IL-17A⁺ cells in CD4⁺CD69⁺ T$_{RM}$ cells isolated from the kidneys of WT$^{MI}$ mice and stimulated with PMA + Iono in the presence of vehicle, 500 μM NaAsO₂, 10 μM Sal003, 20 μM Sephin1 or 20 μM Raphin1 for 3 h ($n = 4$). **f**, Frequency of IL-17A⁺ cells in kidney CD4⁺CD69⁺ T$_{RM}$ cells (left), glomerular crescent score (middle) and urinary albumin to creatinine ratio (right) in *Rag1* KO mice adoptively transferred with eIF2α-S51A⁺ and eIF2α-S51D⁺ T cells ($n = 6$). **g**, Representative histochemistry images of periodic acid-Schiff (PAS)-stained kidney sections from cGN$^{MI}$ mice treated with vehicle ($n = 8$) or raphin1 ($n = 10$) twice a day through oral gavage starting at day 0 of cGN for 10 days until analysis. Arrows indicate glomerular crescents. Scale bars, 50 μm. **h**, Frequency of crescentic glomeruli and the urinary albumin to creatinine ratio in cGN$^{MI}$ mice treated with vehicle ($n = 8$) or raphin1 ($n = 10$) as in **g**. Data are mean + s.e.m. (*$P < 0.05$, **$P < 0.01$). Statistical analysis was performed using one-way ANOVA with Tukey's multiple comparison test (**c**,**e**) or unpaired two-tailed *t*-test with Welch's correction (**a**,**b**,**f**,**h**); *$P < 0.05$, **$P < 0.01$. NS, not significant.

Data Fig. 7b), indicating that CD4⁺CD69⁺ T$_{RM}$ cells were required for the therapeutic effect. These findings suggested that targeting the ISR–eIF2α pathway may represent a therapeutic strategy in immune-mediated diseases.

Finally, to explore the relevance of our findings in human autoimmune diseases, we performed scRNA-seq analysis on CD4⁺ T cells isolated from kidney biopsies of five patients with ANCA-GN

(cohort 3, median age 67 years (IQR 61–72), 60% male). We compared these cells with CD4⁺ T cells from healthy donors (three kidney biopsies from cohort 1, which were processed using the same protocol as the samples from patients with ANCA-GN) (Fig. 5a). The expression levels of mRNA encoding type 1 (for example, *IFNG* and *TNF*) and type 3 (for example, *IL17A* and *IL17F*) cytokines in *S1PR1⁻KLF2⁻*CD69⁺ T$_{RM}$ cells were similar between patients with ANCA-GN and healthy

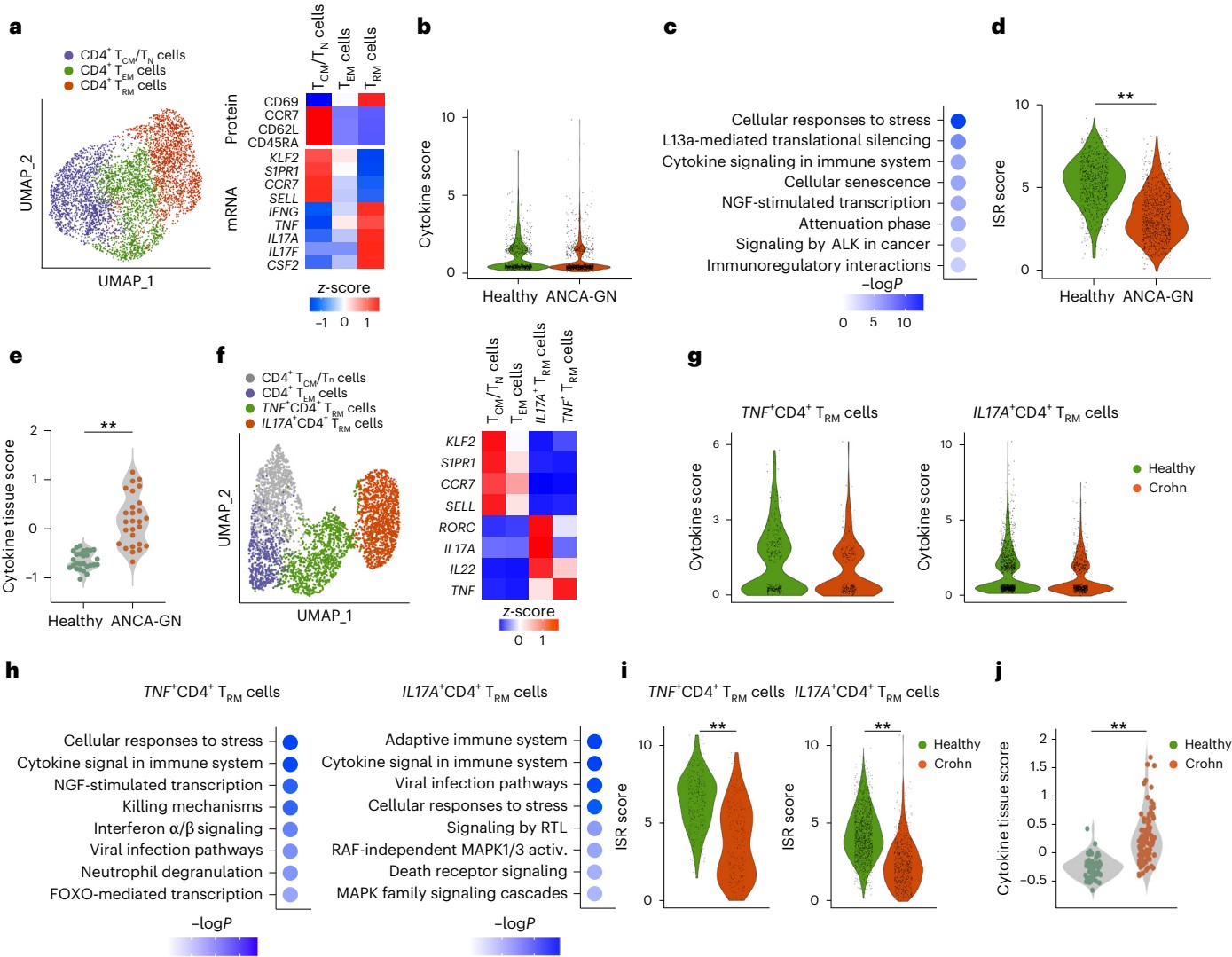

**Fig. 5 | The integrated stress response pathway is downregulated in T_RM cells in patients with inflammatory diseases. a**, UMAP plot showing the distribution of $CD4^+KLF2^-CD69^+$ T_RM cells, $CD4^+SELL^-KLF2^-CD69^-$ T_EM cells and $CD4^+SELL^+KLF2^+$ $CD69^-$ T_CM/T_N cells isolated from the kidney of healthy controls ($n = 3$) and patients with ANCA-GN ($n = 5$) (left) and heatmap showing the expression of CD69, CCR7, CD62L and CD45RA protein and $KLF2$, $S1PR1$, $CCR7$, $SELL$, $IFNG$, $TNF$, $IL17A$, $IL17F$ and $CSF2$ mRNA in the CD4$^+$ T_RM, T_EM and T_CM/T_N cell clusters defined in the UMAP (right). **b**, Violin plot showing cytokine scores based on the expression of $IL2$, $IL17A$, $IL17F$, $IL22$, $IFNG$, $TNF$, $CSF2$ and $LTA$ in kidney CD4$^+$ T_RM cells from healthy controls and patients with ANCA-GN as in **a**. **c**, Reactome analysis showing downregulated pathways in kidney CD4$^+$ T_RM cells from patients with ANCA-GN compared to kidney CD4$^+$ T_RM cells from healthy controls. **d**, Violin plot showing ISR-associated gene scores based on the expression of $PPP1R15A$, $PPP1R15B$, $ATF3$, $ATF4$, $ATF5$, $ATF6$, $DDIT3$, $DNAJB1$, $DUSP1$, $FOS$, $JUN$, $FOSB$ and $JUNB$ in kidney CD4$^+$ T_RM cells from patients with ANCA-GN and healthy controls. **e**, Violin plot showing cytokine-responsive gene scores based on transcriptome data of kidney biopsies from patients with ANCA-GN ($n = 22$) and healthy controls

($n = 21$)[26]. **f**, UMAP plot showing the distribution of $CD4^+S1PR1^-IL17A^{hi}$ T_RM cells, $CD4^+SELL^-S1PR1^-TNF^{hi}$ T_RM cells, $CD4^+SELL^-S1PR1^+$ T_EM cells and $CD4^+SELL^+S1PR1^+$ T_CM/T_N cells isolated from the colon of healthy controls ($n = 2$) and patients with Crohn's disease ($n = 2$) (left) and heatmap showing the expression of $KLF2$, $S1PR1$, $CCR7$, $SELL$, $RORC$, $IL17A$, $IL22$ and $TNF$ mRNA in CD4$^+$ T_CM/T_N, T_EM, $TNF^{hi}$ T_RM and $IL17A^{hi}$ T_RM cell clusters (right). **g**, Violin plot showing cytokine scores in colon CD4$^+TNF^{hi}$ and CD4$^+IL17A^{hi}$ T_RM cells from healthy controls and patients with Crohn's disease as in **f**. **h**, Reactome analysis showing pathways downregulated in colon CD4$^+TNF^{hi}$ and CD4$^+IL17A^{hi}$ T_RM cells from patients with Crohn's disease compared to colon CD4$^+TNF^{hi}$ and CD4$^+IL17A^{hi}$ T_RM cells from healthy controls as in **f**. **i**, Violin plots showing ISR-associated gene scores in colon CD4$^+TNF^{hi}$ and CD4$^+IL17A^{hi}$ T_RM cells from healthy controls and patients with Crohn's disease as in **f**. **j**, Violin plot showing cytokine-responsive gene scores based on transcriptome data of colon biopsies from patients with Crohn's disease ($n = 92$) and healthy controls ($n = 55$). Data are mean + s.e.m. Statistical analysis was performed using unpaired two-tailed $t$-test with Welch's correction (**d**,**e**,**i**,**j**); **$P < 0.01$.

controls (Fig. 5b). Reactome pathway analysis revealed downregulation of the cellular response to stress pathway in kidney CD4$^+$ T_RM cells from patients with ANCA-GN compared to kidney CD4$^+$ T_RM cells from healthy controls (Fig. 5c). Moreover, kidney CD4$^+$ T_RM cells from patients with ANCA-GN exhibited significantly lower expression of ISR-associated genes (for example, $PPP1R15A$, $PPP1R15B$, $ATF3$ and $ATF4$) compared to kidney CD4$^+$ T_RM cells from healthy controls (Fig. 5d and Extended Data Fig. 8a), suggesting CD4$^+$ T_RM cells in patients with ANCA-GN could

have higher mRNA translation efficiency. Of note, cytokine-responsive genes, such as $CXCL1$, $CXCL5$, $CXCL9$ and $CXCL10$ (Extended Data Fig. 3c), were highly expressed in the kidney tissue from patients with ANCA-GN compared to healthy controls (Fig. 5e). These findings suggested that enhanced cytokine mRNA translation contributed to T cell-mediated tissue damage in glomerulonephritis. Similarly, scRNA-seq analysis of CD8$^+$ T cells from these patients found a lower ISR score in kidney CD8$^+$ T_RM cells from patients with ANCA-GN compared to kidney

CD8$^+$ T$_{RM}$ cells from healthy controls (Extended Data Fig. 8b,c), indicating increased CD8$^+$ T$_{RM}$ cell activation in kidneys from patients with ANCA-GN. Additionally, we analyzed a public scRNA-seq dataset of colon CD4$^+$ T cells from healthy donors and patients with Crohn's disease, an inflammatory bowel disease in which type 1 (for example, TNF) and type 3 (for example, IL-17) cytokines play key roles[44]. We identified two CD4$^+$*SELL$^-$CCR7$^-$S1PR1$^-$KLF2$^-$* T$_{RM}$ cell clusters, a *TNF*$^{hi}$ CD4$^+$ T$_{RM}$ cell subset and an *IL17A*$^{hi}$ CD4$^+$ T$_{RM}$ cell subset (Fig. 5f). The expression of mRNA encoding inflammatory cytokines (*IL2*, *IL17A*, *IL17F*, *IL22*, *IFNG*, *TNF*, *CSF2* and *LTA*) in CD4$^+$ T$_{RM}$ cells was comparable between healthy controls and patients with Crohn's disease (Fig. 5g). Both *TNF*$^{hi}$ CD4$^+$ T$_{RM}$ cells and *IL17A*$^{hi}$ CD4$^+$ T$_{RM}$ cells from patients with Crohn's disease exhibited downregulation of the cellular response to stress pathway and lower ISR scores compared to healthy controls (Fig. 5h,i and Extended Data Fig. 8d). In the colon tissue from patients with Crohn's disease, various cytokine-responsive genes were upregulated compared to healthy controls (Fig. 5j), indicating active cytokine mRNA translation in inflamed colon tissue. These findings suggested the potential downregulation of the ISR pathway as a mechanism to enhance the production of effector molecule proteins, resulting in increased tissue inflammation.

In this study, we identified CD4$^+$ T$_{RM}$ cells as the primary T$_M$ cell subset that stored untranslated cytokine mRNA. We showed that the ISR pathway and its mediator p-eIF2α facilitated the mRNA storage thereby suppressing its translation, and that the stored mRNA was sufficient for the rapid cytokine production upon activation. Different mechanisms of translation regulation have been reported. How the ISR pathway synergizes with other mechanisms, such as RNA-binding proteins[21,45], the mTOR pathway[46] or regulation of mRNA half-life[21,47], remains unclear. While enhancing ISR with chemical compounds or gene overexpression significantly reduced cytokine production in mouse kidney CD4$^+$ T$_{RM}$ cells, the marginal effects of ISRIB on cytokine production in these cells suggested that other pathways also regulate cytokine mRNA translation in CD4$^+$ T$_{RM}$ cells. Overall, our findings suggested that the ISR–eIF2α pathway regulates cytokine mRNA translation in CD4$^+$ T$_{RM}$ cells during homeostasis, and identified this pathway as a potential therapeutic target in T cell-mediated inflammatory diseases where ISR-mediated regulation is impaired.

## Online content

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

[1]III. Department of Medicine, University Medical Center Hamburg-Eppendorf, Hamburg, Germany. [2]Hamburg Center for Translational Immunology, University Medical Center Hamburg-Eppendorf, Hamburg, Germany. [3]Institute of Medical Systems Biology, Center for Biomedical AI, Center for Molecular Neurobiology Hamburg, Hamburg, Germany. [4]Department of Urology, University Medical Center Hamburg-Eppendorf, Hamburg, Germany. [5]The Calcium Signalling Group, Department of Biochemistry and Molecular Cell Biology, University Medical Center Hamburg-Eppendorf, Hamburg, Germany. [6]Institute of Biochemistry and Molecular Biology, University of Hamburg, Hamburg, Germany. [7]Hamburg Center for Kidney Health (HCKH), University Medical Center Hamburg-Eppendorf, Hamburg, Germany. [8]Institute of Systems Immunology, University Medical Center Hamburg-Eppendorf, Hamburg, Germany. [9]Department of General, Visceral and Thoracic Surgery, University Medical Center Hamburg-Eppendorf, Hamburg, Germany. [10]I. Department of Medicine, University Medical Center Hamburg-Eppendorf, Hamburg, Germany. [11]Institute for Immunology, University Medical Center Hamburg-Eppendorf, Hamburg, Germany. [12]These authors jointly supervised this work: Hans-Willi Mittrücker, Christian F. Krebs, and Ulf Panzer. ✉e-mail: panzer@uke.uni-hamburg.de

## Methods

### Human studies

Human kidney tissue was obtained from $n$ = 8 donors (included in the cohort 1) and $n$ = 5 donors (included in the cohort 2) registered to the Hamburg GN Registry; $n$ = 21 healthy donors, $n$ = 22 patients with ANCA-GN, $n$ = 32 patients with lupus nephritis and $n$ = 27 patients with IgA nephropathy registered to the European Renal cDNA Bank[26]; and from $n$ = 5 patients (included in cohort 3) registered to the Clinical Research Unit 228 (CRU 228) ANCA-GN cohort. For the eight donors in cohort 1 and the five donors in cohort 2, matched blood samples from the same donors were also analyzed for this study. Kidney T cells isolated from the healthy part of kidneys removed as part of tumor nephrectomy were considered as T cells at steady state. The histological examination confirmed that the healthy part of the kidney did not exhibit any signs of tumor invasion or active immune cell invasion. The research studies were conducted with the approval of the Ethik-Kommission der Ärztekammer Hamburg (the local ethics committee of the Chamber of Physicians in Hamburg) and in compliance with the ethical principles outlined in the Declaration of Helsinki. All participating patients gave informed consent for the research.

### Mouse experiments

Experiments with mice followed the national guidelines, and local ethics committees (Behörde für Justiz und Verbraucherschutz Hamburg) approved the research protocols. Mice were housed under specific-pathogen-free conditions to ensure the validity and reliability of the experiments. Age-matched C57BL/6 male mice (between 8 and 16 weeks of age) were used in all experiments to minimize the variability between subjects. $Il17a^{Cre}R26^{eYFP}$ male mice[32] between 8 and 16 weeks of age were used to identify $T_{RM}17$ cells. For the induction of IL-17A^{MI} and WT^{MI} mice models, $Il17a^{Cre}R26^{eYFP}$ or wild-type C57BL/6 mice were intravenously injected with $1 \times 10^7$ $S.$ $aureus$ (strain SH1000[13]) in 100 µl sterile PBS. For pathogen and inflammation clearance, ampicillin was given in drinking water for 2 weeks[13] and mice were analyzed 2 months after the completion of the antibiotic treatment. Experimental crescentic glomerulonephritis was induced by intraperitoneal administration of sheep nephrotoxic serum targeting the glomerular basement membrane[13]. The cGN^{MI} mice model was induced by administration of the sheep nephrotoxic serum into WT^{MI} mice models. $R$aphin1 was reconstituted in distilled water with 0.5% methylcellulose (Sigma, M0512) and administered twice a day by oral gavage.

### Histopathology, immunohistochemistry and immunofluorescence

To evaluate glomerular crescent formation, 30 glomeruli per mouse were assessed in a blinded manner in PAS-stained paraffin sections of kidneys[13]. For immunofluorescence staining, primary antibodies, including CD3 (A0452, Dako), Phospho-eIF2α (MA5-32021, Invitrogen), total eIF2α (D7D3, Cell Signaling) and Tia1 (ab140595, Abcam) were incubated in blocking buffer at 4 °C for 1 h. Following PBS washing, fluorochrome-labeled secondary antibodies were applied, and the staining was visualized using an LSM800 with Airyscan and the ZenBlue software (all Carl Zeiss).

To detect mRNA (FISH) in mice and human kidney sections, RNAscope Multiplex Fluorescent Assay (Advanced Cell Diagnostics) was employed. mRNA detection (FISH) in sorted T cells was carried out using the ViewRNA Cell Plus Assay kit (cat. no. 88-19000, Invitrogen). The slides were imaged using a Zeiss LSM800 confocal microscope and analyzed with ZEN software (Carl Zeiss). To evaluate $Il17a$ mRNA in stress granules, the percentage of $Il17a$ signals overlapping with Tia1-positive granules was calculated.

### Isolation and flow cytometric analysis of human and murine leukocytes

Single-cell suspensions were obtained from kidney and blood samples to isolate and analyze human leukocytes. Kidney tissue was enzymatically digested with collagenase D at 0.4 mg ml$^{-1}$ (Roche) and DNase I (10 µg ml$^{-1}$, Sigma-Aldrich) in RPMI 1640 medium at 37 °C for 30 min, followed by dissociation with gentleMACS (Miltenyi Biotec). Blood samples were separated using Leucosep tubes (Greiner Bio-One). Samples were filtered through a 30-µm filter (Partec) before antibody staining and flow cytometry.

Cells from murine spleens were isolated by squashing the organ through a 70-µm cell strainer. Erythrocytes were lysed using a lysis buffer (155 mM $NH_4Cl$, 10 mM $KHCO_3$ and 10 µM EDTA, pH 7.2). To isolate renal lymphocytes from mice, kidneys were enzymatically digested with 400 µg ml$^{-1}$ collagenase D (Roche) and 10 U ml$^{-1}$ DNase I (Sigma-Aldrich) at 37 °C for 30 min. Subsequently, leukocytes were isolated by density gradient centrifugation using 37% Easycoll (Merck Millipore) and a filtration step using a 30-µm cell strainer (Partec). T cell isolation from the intestine is described previously. In brief, murine intestine was cut longitudinally after removing Peyer's patches and adventitial fat. To collect intraepithelial lymphocytes, the intestine tissue was incubated in HBSS containing 1 mM dithiothreitol (DTT) followed by a dissociation step using 1 mM EDTA for 20 min at 37 °C. To collect lamina propria lymphocytes, the tissue was cut into small pieces and incubated for 45 min at 37 °C in HBSS supplied with 1 mg ml$^{-1}$ collagenase and 10 U ml$^{-1}$ DNase I. Leukocytes were further enriched by Percoll gradient. Intraepithelial cells and lamina propria lymphocytes were pooled for analysis.

Cells were surface stained with fluorochrome-conjugated antibodies (human, CD45 (HI30, BD Biosciences), CD3 (OKT3, eBioscience), CD4 (RPA-T4, BD Biosciences), CD8 (RPA-T8, BD Biosciences), CD69 (FN50, BD Biosciences), CD45RA (HI100, BD Biosciences), GM-CSF (VD2-21C11, BD Biosciences), IL-17A (BL168, BioLegend), IL-17F (SHLR17, BioLegend), IFNγ (4S.B3, eBioscience), TNF (MAB11, BD Biosciences), CCR7/CD197 (Go43H7, BD Biosciences); mouse, CD45 (30-F11, BD Biosciences), CD3 (145-2C11, BD Biosciences), CD4 (RM4-5, BD Biosciences), CD8 (53-6.7, BD Biosciences), GM-CSF (MP1-22E9, BD Biosciences), IL-17A (TC11-18H10, BD Biosciences), IL-17F (9D3.1C8, BioLegend), IL-22 (poly5164, BioLegend), IFNγ (XMG1.2, BD Biosciences), TNF (MP6-XT22, BD Biosciences)) and a fixable dead-cell stain (Molecular Probes) to exclude dead cells from the analysis. For intracellular staining, samples were processed using Cytofix/Cytoperm (BD Biosciences) according to the manufacturer's instructions. All antibodies were diluted at a ratio of 1:100 to 1:200.

To assess the phospho-eIF2α and total eIF2α levels, isolated T cells were stimulated with CD3/28 antibodies for 2–3 days, and transferred to unstimulated conditions. For re-stimulation, T cells were further stimulated with PMA + Iono or CD3 + CD28 antibodies. Stimulated and unstimulated T cells underwent surface staining and were subsequently fixed and permeabilized using Foxp3/Transcription Factor Staining Buffer (eBioscience). The cells were then incubated with primary antibodies directed against phospho-eIF2α or total eIF2α (1:1,000 dilution) for 40 min. After two rounds of washing, the cells were exposed to a secondary antibody conjugated with AF647 against Rabbit IgG (1:1,000 dilution) for 40 min.

### Ca^{2+} live-cell imaging in murine T cells

Freshly isolated primary murine CD4$^+$ T cells were loaded with Fura2-AM (4 µM) and incubated for 40 min at 37 °C. Following the loading process, the cells were washed twice and resuspended in Ca$^{2+}$ buffer comprising 140 mM sodium chloride (NaCl), 5 mM potassium chloride (KCl), 1 mM magnesium sulfate ($MgSO_4$), 1 mM calcium chloride ($CaCl_2$), 20 mM 4-(2-hydroxyethyl)-1 piperazine ethanesulfonic acid (HEPES), 1 mM sodium dihydrogen phosphate ($NaH_2PO_4$) and 5.5 mM glucose at pH 7.4. Imaging was conducted using a Leica IRBE microscope with a ×40 magnification and an exposure of 25 ms. The microscope was equipped with a Sutter DG-4 as a light source and an electron-multiplying charge-coupled device camera (EMCCD; 13, Hamamatsu). The images were acquired in 16-bit mode with an

acquisition rate of one frame every 2 s, using Volocity software (v.6.6.2; PerkinElmer). The following filter set for Fura2 was used (excitation filters, HC 340/26, HC 387/11; beamsplitter, 400 DCLP; emission filter, 510/84, all in nanometers). At 5 min before acquisition, the T cells were incubated with either 10 µM Sal003 (Sigma-Aldrich, S4451), 20 µM Sephin1 (Sigma-Aldrich, SML1356), 20 µM Raphin1 (Sigma-Aldrich, SML2562), 500 µM NaAsO$_2$ (Sigma-Aldrich, S7400) or Ca$^{2+}$ buffer and dimethylsulfoxide (DMSO) (as control). T cell stimulation was initiated through the addition of soluble anti-CD3 antibody (10 µg ml$^{-1}$) after 1 min. Subsequently, the background correction, splitting of fluorescence channels and selection of cells into regions of interest were performed with Fiji v.2. Furthermore, to obtain the Ca$^{2+}$ concentration from ratio values, a calibration was calculated by measuring the maximal ratio value (Rmax) and the minimal ratio value (Rmin). The mean Ca$^{2+}$ concentration over time as well as the mean basal (after 30 s), peak (at 260 s) and plateau (at 520 s) Ca$^{2+}$ concentration were calculated using Prism10.

### Flow cytometry and cell sorting
Samples were measured with LSR II or Symphony A3 (both BD Biosciences). Data analysis was performed using FlowJo software (TreeStar). FACS sorting was performed with an AriaFusion or AriaIIIu (BD Biosciences).

### Quantitative RT–PCR
T cells were subjected to RNA extraction utilizing the RNeasy Micro kit (QIAGEN). RNA from the renal cortex was isolated with the NucleoSpin kit (Macharey-Nagel) in compliance with the manufacturer's recommended protocol. Subsequently, the RNA underwent reverse transcription using the High-Capacity cDNA Reverse Transcription kit (Thermo Fisher), and the StepOnePlus Real-Time PCR system (Thermo Fisher) was utilized for measurement. The Taqman primers used for human *IL17A*, *IFNG*, *CSF2*, *ZC3H12D*, *ATF4*, *18S rRNA* and murine *Il17a* were procured from Life Technologies.

### In vitro stimulation of cells
Human and murine T cells were classified based on the surface markers and FACS sorted before stimulation. T$_{RM}$ cells were characterized as CD45$^+$CD4$^+$CD44$^+$CD69$^+$CD62L$^-$; T$_{EM}$ cells as CD45$^+$CD4$^+$CD44$^+$CD69$^-$CD62L$^-$; T$_{CM}$ cells as CD45$^+$CD4$^+$CD44$^+$CD69$^-$CD62L$^+$; and naive T cells as CD45$^+$CD4$^+$CD44$^-$CD69$^-$CD62L$^+$. The cells were exposed to PMA (50 ng ml$^{-1}$, Sigma-Aldrich) and Iono (1 µM, Sigma-Aldrich) for T cell activation. To detect cytokine production, brefeldin A (10 µg ml$^{-1}$, Sigma-Aldrich) was added to inhibit cytokine secretion from cells. After 3–6 h, cytokine production was measured using intracellular staining and flow cytometry techniques. To inhibit de novo mRNA transcription, 20 µg ml$^{-1}$ actinomycin D (A1410, Sigma-Aldrich) was added to the medium 10 min before T cell stimulation. For ex vivo stimulation to measure eIF2α levels, 1–2 × 10$^6$ T$_{RM}$ cells were cultured in a volume of 300 µl IMDM containing 10% FCS, streptomycin and penicillin in a 96-well plate pre-coated with anti-CD3 antibody (2 µg ml$^{-1}$, BioLegend 100360), along with the addition of 5 µg ml$^{-1}$ anti-CD28 antibody (BioLegend 102122) for a duration of 2 days. To induce the ISR, T cells were incubated with 10 µM Sal003 (Sigma-Aldrich, S4451), 20 µM Sephin1 (Sigma-Aldrich, SML1356), 20 µM Raphin1 (Sigma-Aldrich, SML2562) or 500 µM NaAsO$_2$ (Sigma-Aldrich, S7400). For the experiment using ISRIB, ex vivo-cultured T$_{RM}$ cells isolated from the kidney were treated with 500 nM ISRIB (Sigma, SML0843) overnight and analyzed for the production of cytokines.

### Puromycin incorporation assay
T cells were cultured in the presence or absence of PMA (50 ng ml$^{-1}$, Sigma-Aldrich) and Iono (1 µM, Sigma-Aldrich) for 1 h. Puromycin (5 µg ml$^{-1}$, Sigma-Aldrich) was added in the last 10 min. Cells were washed and stained for surface markers and fixable dead-cell stain and then fixed and permeabilized using Foxp3/Transcription Factor Staining Buffer (eBioscience). Puromycin was detected using anti-puromycin-AF647 (MABE343 clone 12D10, Sigma-Aldrich).

### Polysome profiling for mRNA translation efficiency analysis
Polysome profiling and mRNA extraction were conducted as described previously[30]. Human T$_{RM}$ cells sorted from healthy human kidney tissue were mixed with two million NIH/3T3 cells because the number of human T$_{RM}$ cells was insufficient to show monosome peaks. We relied on monosome peaks from a murine cell line because monosome peaks were observed with the same kinetics in humans and mice. Cells were lysed in 250 µl cell lysis buffer (10 mM Tris-HCl (pH 7.4), 5 mM MgCl$_2$, 100 mM KCl, 1% Triton X-100, 2 mM DTT and 100 µg ml$^{-1}$ cycloheximide). Cells were shear-opened with a 26-gauge needle by passing the lysate eight times through it. A quantity of 300 µl of the cell lysate was loaded onto a 5-ml sucrose gradient (60% to 15% sucrose) dissolved in polysome buffer (50 mM HEPES-KOH, pH 7.4, 5 mM MgCl$_2$, 100 mM KCl, 2 mM cycloheximide and 2 mM DTT) and separated by ultracentrifugation at 148,900$g$ (Ti55 rotor, Beckman) for 1.5 h at 4 °C. Polysome fractions and nonpolysome fractions were collected separately. RNA was extracted from each fraction by adding 0.1 volume of 10% SDS and one volume of acidic phenol–chloroform (5:1, pH 4.5), incubated at 65 °C for 5 min, and centrifuged at 21,000$g$ for 5 min at 4 °C to separate different phases. Equal volumes of acid phenol–chloroform were added to the aqueous phase, separated by centrifugation and supplemented with an equal volume of chloroform:isoamyl alcohol (24:1). Upon separation, the aqueous phase was supplemented with 0.1 volume 3 M NaOAc (pH 5.5) and an equal volume of isopropanol, precipitated at −20 °C for 3 h. RNA was pelleted at 21,000$g$ at 4 °C, and the dried pellets were resuspended in water. RNA was subjected to RT–qPCR analysis and the original mRNA amount in polysome and nonpolysome fractions was calculated by RT–qPCR using Taqman primers specific to human mRNA. mRNA translation efficiency was defined as the amount of mRNA in polysome fraction divided by the total mRNA amount of nonpolysome and polysome fractions.

### Retrovirus transduction and cell transfer into *Rag1* KO mice
MIGR1 plasmid[48] (produced by W. Pear, Addgene plasmid #27490) was used to overexpress eIF2α variants. Both eIF2α variants, S51A and S51D, were inserted into EcoRI-digested MIGR1 backbone. To produce retrovirus, HEK293T cells were seeded into a six-well plate and, the next day, transfected with MIGR1-eIF2α variant plasmid and pCL-Eco plasmid[49] (Produced by Inder Verma, Addgene plasmid #12371) using Lipofectamine 3000 (Invitrogen L3000001). Murine T cells were stimulated in a 24-well plate (2 × 10$^6$ cells per well) for 2 days using anti-CD3 antibody (2 µg ml$^{-1}$, BioLegend, 100360) and anti-CD28 antibody (5 µg ml$^{-1}$, BioLegend, 102122). After the 2-day stimulation, the medium was replaced with virus-containing medium with supplementation of Polybrene. Cells were centrifuged at 1,000$g$, room temperature for 1 h. The next day, green fluorescent protein-positive T cells were sorted using FACS and transferred into *Rag1* KO mice (10$^5$ cells per mouse). Following T cell reconstitution, mice were challenged by the cGN model.

### Single-cell RNA sequencing
To carry out scRNA-seq, single-cell suspensions were derived from human and mouse kidney samples. Cell hashing was implemented according to the manufacturer´s instructions (BioLegend). For CITE-seq, monoclonal antibodies with corresponding barcodes were applied to samples (BioLegend). FACS-sorted CD45$^+$ cells or CD3$^+$ T cells were subjected to droplet-based single-cell analysis and transcriptome library preparation using Chromium Single-Cell 5′ Reagent kits v2 according to the manufacturer's protocols (10x Genomics). The generated scRNA-seq libraries were subjected to sequencing on a NovaSeq6000 system (100 cycles) (Illumina).

# Letter

## Alignment, quality control and pre-processing of scRNA-seq data

Quality control and scRNA-seq pre-processing were carried out as previously described[13]. In brief, the Cell Ranger software pipeline (v.5.0.1, 10x Genomics) was utilized to perform the demultiplexing of cellular barcodes and mapping of reads to the reference genome (refdata-cellranger-hg19-1.2.0 (human) or refdata-gex-mm10-2020-A (mouse)). Seurat (v.4.0.2) demultiplexing function HTODemux was used to demultiplex the hash-tag samples. We filtered out the cells with <500 genes, >6,000 genes or >5% mitochondrial genes. For CITE-seq, a pseudo-reference genome was built with cellranger mkref function in Cell Ranger. CITE-seq raw data were aligned to this pseudo-reference genome using Cell Ranger function cellranger count. The antibody-derived tags data were integrated to scRNA-seq data using Seurat. Further information is described previously[13].

## Dimensionality reduction, clustering, enrichment analysis and scores

The Seurat package (v.4.0.2) was used to conduct unsupervised clustering analysis on scRNA-seq data. In brief, gene counts for cells were normalized by library size and log-transformed. To reduce batch effects, we employed the integration method implemented in the latest Seurat v.4 (function FindIntegrationAnchors and IntegrateData, dims of 1:30). The integrated matrix was then scaled by the ScaleData function (default parameters). To reduce dimensionality, principal-component analysis was performed on the scaled data (function RunPCA, npcs of 30). Thirty principal components were determined using the ElbowPlot function to compute the KNN graph based on the Euclidean distance (function FindNeighbors), which then generated cell clusters using the function FindClusters. UMAP was used to visualize clustering results. The top differential expressed genes in each cluster were found using the FindAllMarkers function (min.pct of 0.1) and running Wilcoxon rank-sum tests. The differential expression between clusters or groups was calculated by the FindMarkers function (min.pct of 0.1), which also included Wilcoxon rank-sum tests. For enrichment analysis including GO, Reactome, Canonical pathway and protein–protein interaction analyses, metascape[50] was used. To calculate scores (RPG score and ISR score) in scRNA-seq data, Seurat function AddModuleScore was used. For both scores, all RPGs[34] and ISR-associated genes[31,38–41] listed in heatmaps (Extended Data Fig. 5) were included.

## Data analysis, statistics and reproducibility

Statistical analysis was performed using GraphPad Prism. The results are shown as mean ± s.e.m. when presented as a bar graph or as single data points with the mean in a scatter-plot. Differences between the two individual groups were compared using a two-tailed $t$-test. In the case of three or more groups, a one-way ANOVA with Tukey's multiple comparisons test was used. The correlation coefficient $r$ was calculated using a Pearson correlation, and the corresponding $P$ value was based on a $t$-distribution test. All experiments were repeated at least three times, except for some experiments with human cells, which were repeated twice as stated in the figure legends. No statistical methods were used to predetermine sample sizes but our sample sizes are similar to those reported in previous publications[13]. Data distribution was assumed to be normal but this was not formally tested. Data collection was not performed but data analysis was performed blind to the conditions of the experiments. We did not exclude any data points. For bioinformatics analysis, RStudio and Jupyter Notebook were used to execute code.

## Reporting summary

Further information on research design is available in the Nature Portfolio Reporting Summary linked to this article.

## Data availability

scRNA-seq data of T cells from human healthy kidney tissue and $T_{RM}$ cell-induced murine kidneys are available at the Gene Expression Omnibus (GEO) under accession code GSE270533 (ref.13) and figshare (https://figshare.com/s/7912de1afc7fd5bbefd4). The scRNA-seq data of the healthy colon and lung were obtained from GEO under GSE157477 (ref. 24) and Cross-Tissue Immune Cell Atlas (https://www.tissue-immunecellatlas.org/#publication)[25], respectively. The transcriptome data of European Renal cDNA Bank (kidney biopsy transcriptome datasets) are available from GEO at GSE104948 (ref. 26) and refine.bio (https://www.refine.bio/experiments/GSE104948/glomerular-transcriptome-from-european-renal-cdna-bank-subjects-and-living-donors). The transcriptome datasets of colon tissues from patients with Crohn's disease and healthy individuals are available from GEO at GSE109142 (ref. 24). For the reference genome, refdata-cellranger-hg19-1.2.0 (human) and refdata-gex-mm10-2020-A (mouse) were used. All other data needed to verify the study's conclusions are contained in the paper or the Supplementary Materials.

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

## Acknowledgements

FACS sorting was performed at the UKE FACS sorting core facility. scRNA-seq was performed at the UKE Single-Cell Core Facility. Graphical images were produced using Servier Medical Art images (https://smart.servier.com) and TogoTV (https://togotv.dbcls.jp). This study was supported by grants from the Deutsche Forschungsgemeinschaft to U.P. (SFB 1192 A1 and C3), C.F.K. (SFB 1192 A5 and C3; KR 3483/3-1) and H.-W.M. (SFB 1192 A4). N.A. was supported by a Research Fellowship of the Japan Society for the Promotion of Science. The funders had no role in study design, data collection and analysis, decision to publish or preparation of the paper.

## Author contributions

Conceptualization: N.A. and U.P. Methodology: N.A., P.G., H.-J.P., N.S., B.-P.D., A.P., A.K., H.W. and N.G. Formal analysis: N.A., C.F.K. and U.P. Flow cytometry: N.A., P.G. and H.-J.P. scRNA sequencing data analysis: N.A., P.G. and Y.Z. Data analysis: N.A., P.G., R.K., B.-P.D., A.K. and A.P. Polysome profiling: N.A., S.D. and Z.I. Renal histology: N.A. and U.P. Patient cohorts: J.E., J.-H.R., P.G., R.D., C.F.K. and U.P. Writing—original draft: N.A. and U.P. Writing—review and editing: N.A., H.-W.M., C.F.K. and U.P. Visualization: N.A. and U.P. Supervision: J.-E.T., Z.I., T.B.H., I.P., N.G., H.-W.M., C.F.K. and U.P. Funding acquisition: H.-W.M., C.F.K. and U.P.

## FundingInformation

## Competing interests

The authors declare no competing interests.

## Additional information

**Extended data** is available for this paper at https://doi.org/10.1038/s41590-025-02105-x.

**Correspondence and requests for materials** should be addressed to Ulf Panzer.

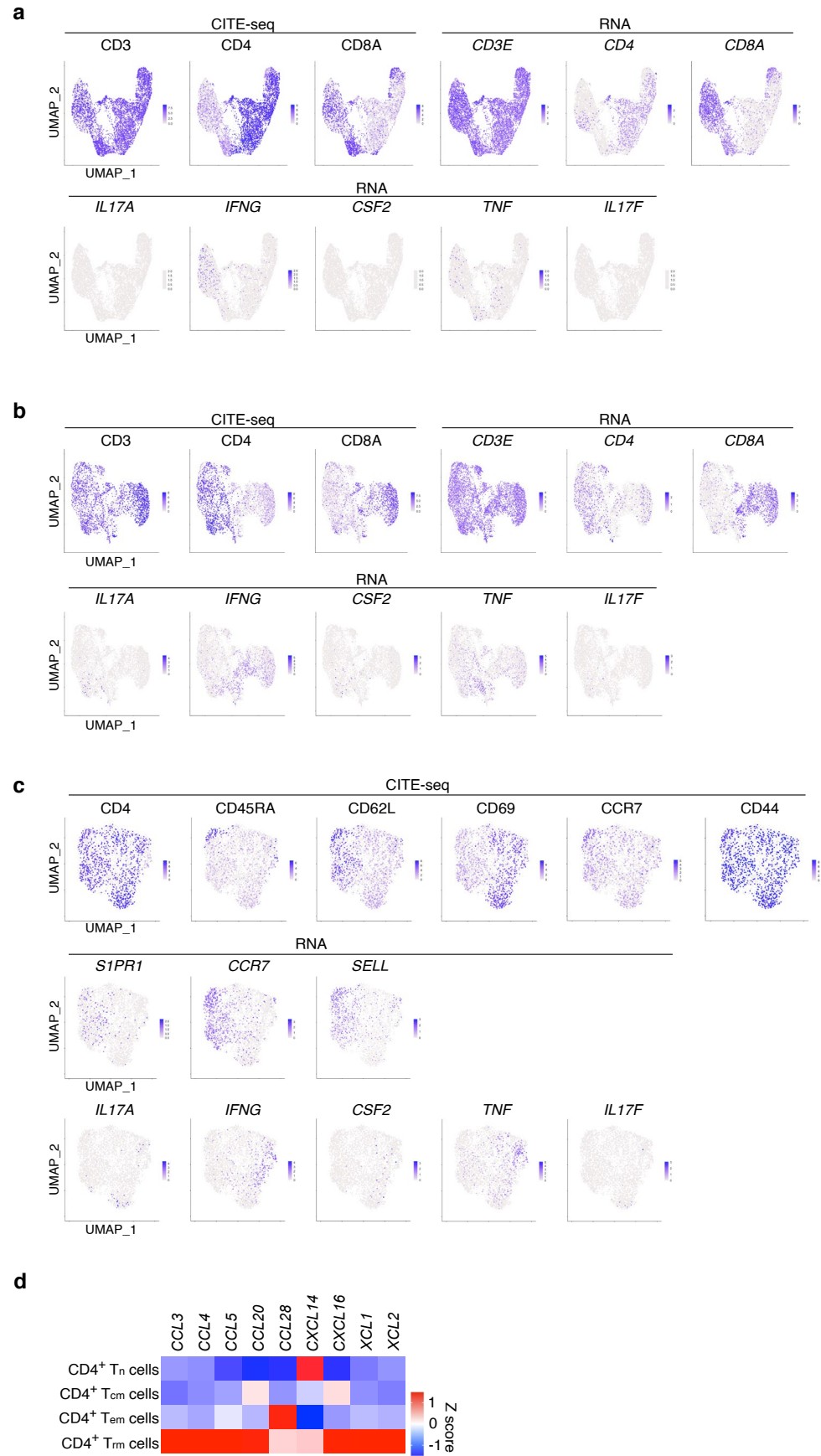

**Extended Data Fig. 1 | See next page for caption.**

**Extended Data Fig. 1 | scRNA-seq analysis of T cells from human healthy kidney tissue and matched blood. a,b,** UMAP plots showing the expression of marker protein (CD3, CD4 and CD8A) and mRNA (*CD3E*, *CD4* and *CD8A*) as well as cytokine mRNA (*IL17A*, *IFNG*, *CSF2*, *TNF* and *IL17F*) in CD4+ and CD8+ T cells in the matched blood (**a**) and the healthy kidney (**b**). Data are produced by scCITE-seq analysis. **c**, UMAP plots showing the expression of marker protein (CD4, CD45RA,

CD62L, CD69, CCR7 and CD44) and mRNA (*S1PR1*, *CCR7* and *SELL*) as well as cytokine mRNA (*IL17A*, *IFNG*, *CSF2*, *TNF* and *IL17F*) in CD4+ T cells isolated from the human healthy kidney. Data are produced by scCITE-seq analysis of CD4+ T cell subset data. **d**, Heatmap showing the expression of chemokine genes (*CCL3*, *CCL4*, *CCL5*, *CCL20*, *CCL28*, *CXCL14*, *CXCL16*, *XCL1* and *XCL2*) in CD4+ $T_{rm}$, $T_{em}$, $T_{cm}$ and $T_n$ cell clusters.

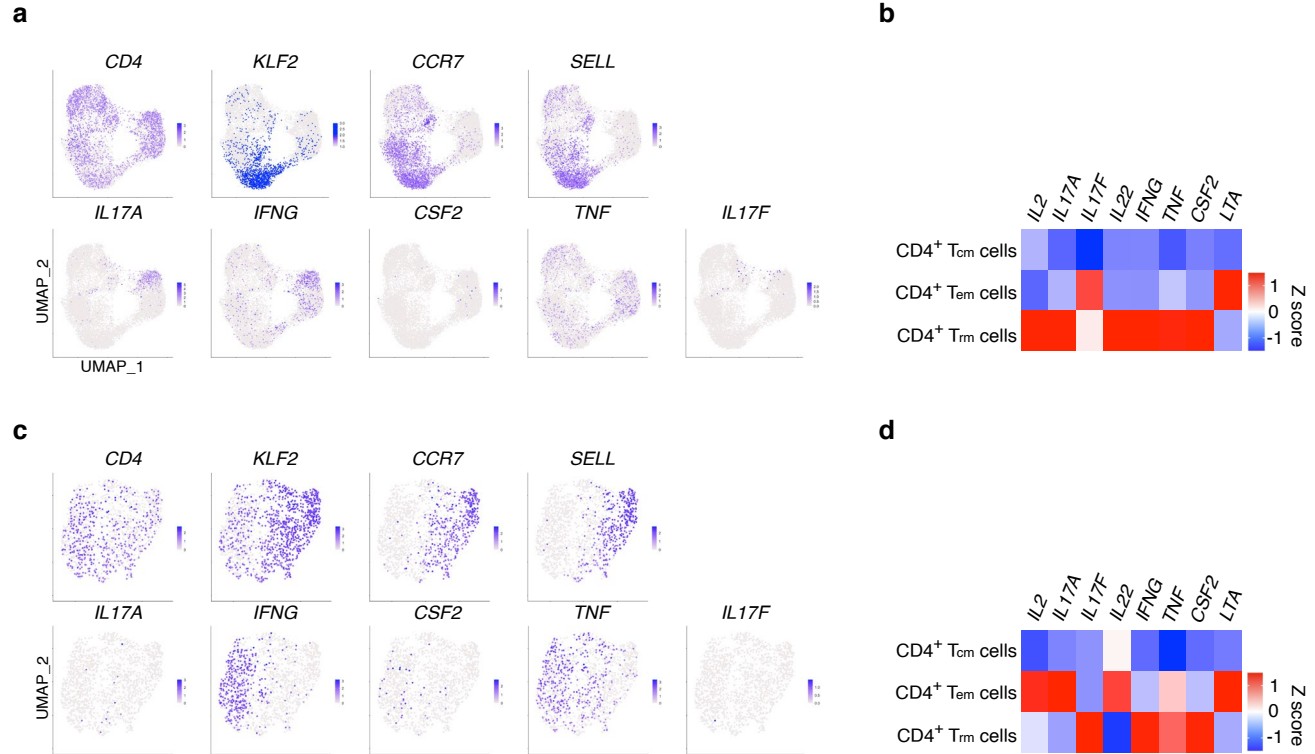

**Extended Data Fig. 2 | Cytokine mRNA expression in the CD4$^+$ Trm cells from human healthy colon and lung. a,b,** UMAP plots showing the expression of marker genes (*CD4*, *KLF2*, *CCR7* and *SELL*) and cytokine mRNA (*IL17A*, *IFNG*, *CSF2*, *TNF* and *IL17F*) (**a**) and a heatmap showing cytokine mRNA expression (*IL2*, *IL17A*, *IL17F*, *IL22*, *IFNG*, *TNF*, *CSF2* and *LTA*) (**b**) in CD4$^+$ T$_{rm}$, T$_{em}$ and T$_{cm}$ cell clusters from the heatlhy colon. **c,d,** UMAP plots showing the expression of marker genes (*CD4*, *KLF2*, *CCR7* and *SELL*) and cytokine mRNA (*IL17A*, *IFNG*, *CSF2*, *TNF* and *IL17F*) (**c**) and a heatmap showing cytokine mRNA expression (*IL2*, *IL17A*, *IL17F*, *IL22*, *IFNG*, *TNF*, *CSF2* and *LTA*) (**d**) in CD4$^+$ T$_{rm}$, T$_{em}$ and T$_{cm}$ cell clusters from the heatlhy lung.

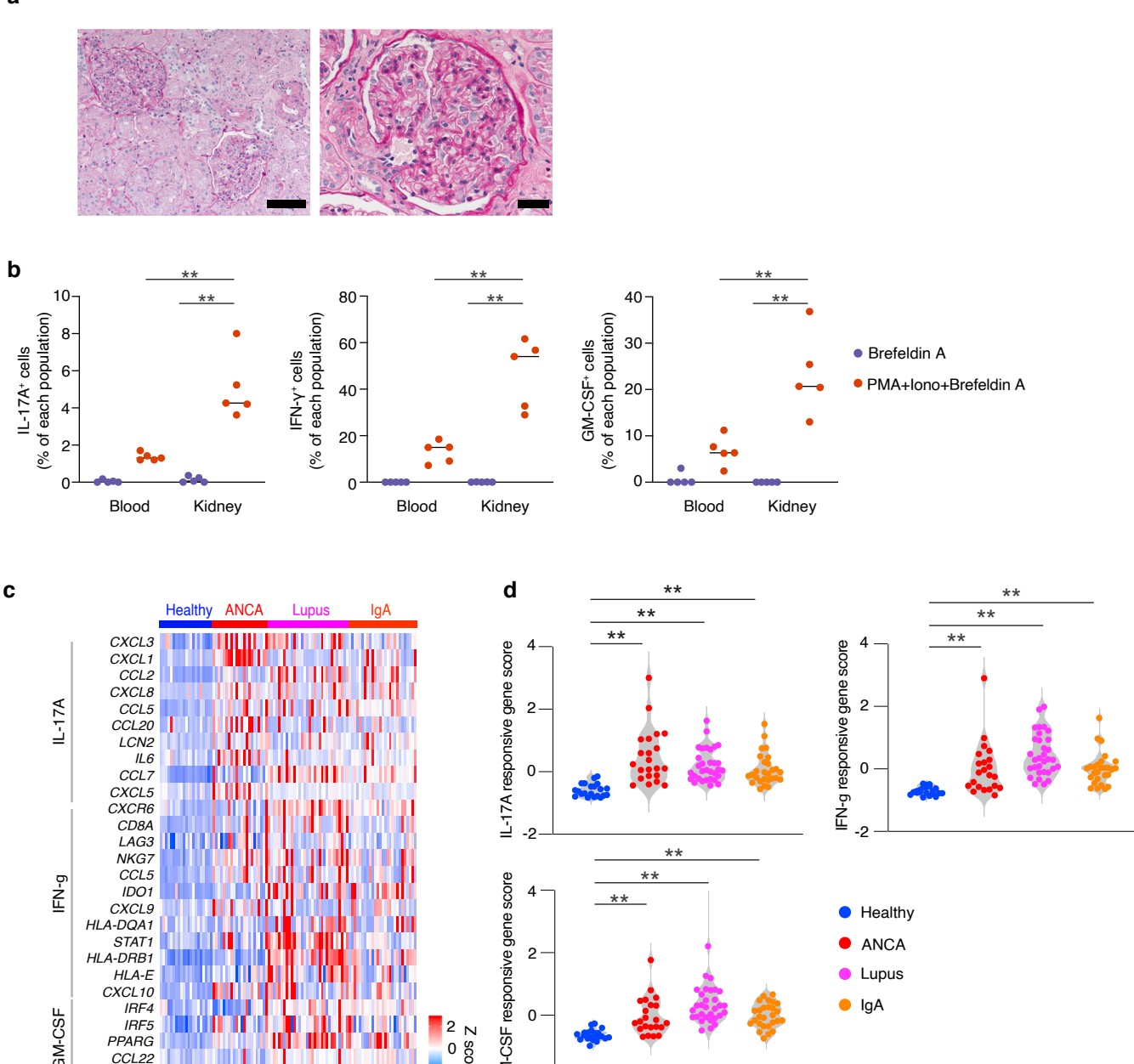

**Extended Data Fig. 3 | Inflammatory response is not detected in healthy human kidney tissue. a**, Representative images showing PAS-stained human healthy kidney sections. Scale bars indicate 100 mm (left) and 40 mm (right). The experiment was repeated 5 times. **b**, Graphs showing the frequency of IL-17A⁺, IFN-γ⁺ and GM-CSF⁺ CD4⁺ T cells isolated from the matched blood and healthy kidney. T cells were treated with Brefeldin A in the presence or absence of PMA+Iono for 6 hours (n = 5). **c,d**, Heatmap showing the expression of cytokine-responsive genes in the kidney (**c**) and graphs showing the quantification of the cytokine-responsive genes as a score (Healthy: n = 21; ANCA: n = 22; Lupus: n = 32; IgA: n = 27) (**d**). Statistical analysis was performed using one-way ANOVA with Tukey's multiple comparison test; * $p < 0.05$, ** $p < 0.01$.

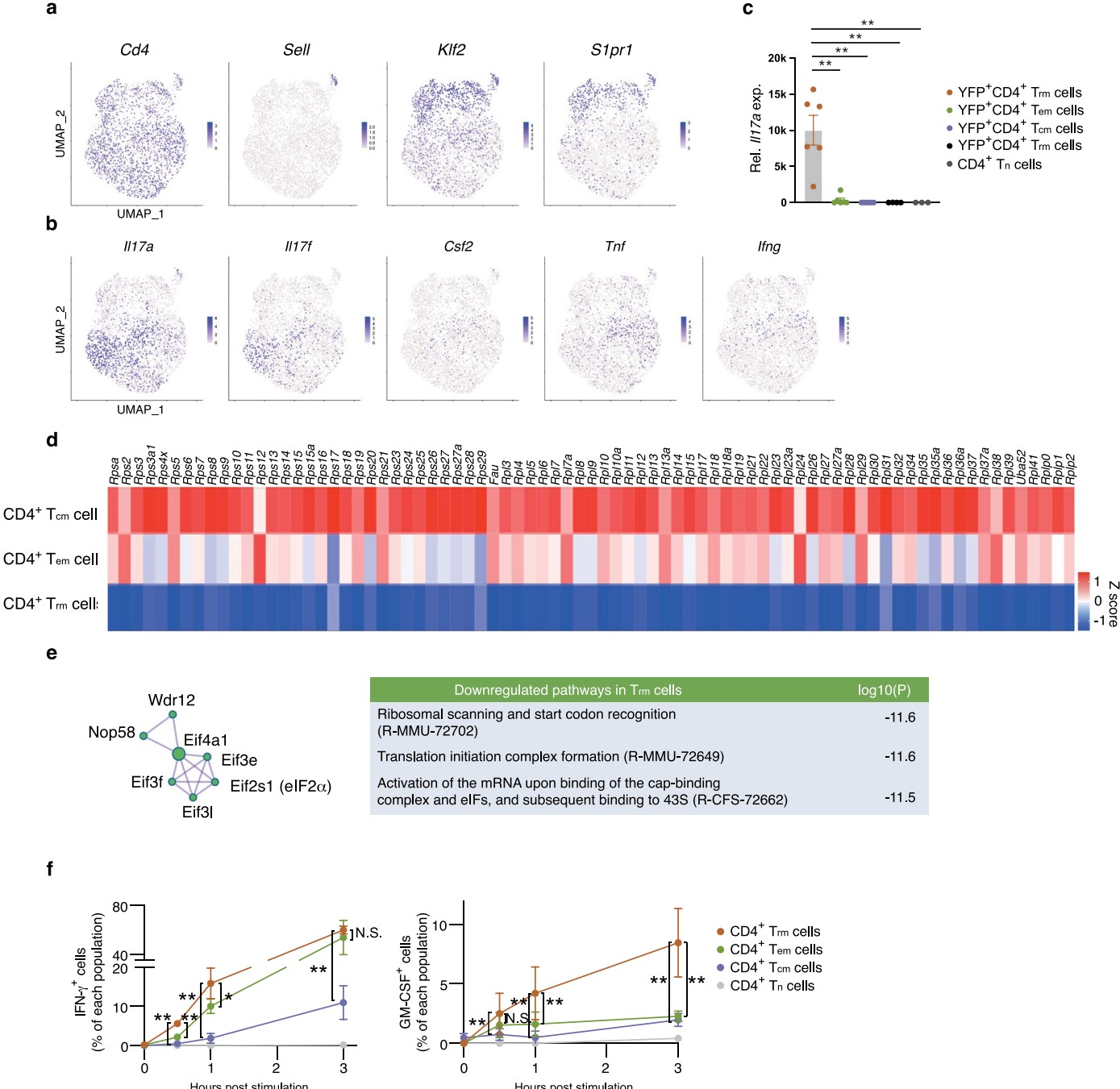

**Extended Data Fig. 4 | Murine renal CD4⁺ T$_{rm}$ cells express high levels of cytokine mRNA and produce cytokine protein immediately after activation. a,b,** UMAP plots showing the expression of marker genes (**a**) and proinflammatory cytokine (**b**) in YFP⁺CD4⁺ T cells isolated from the kidneys of IL-17A$^{MI}$ mice analyzed with scRNA-seq. **c,** Real-time RT−PCR analysis showing the levels of *Il17a* mRNA in YFP⁺CD4⁺CD44⁺CD69⁺ T$_{rm}$ cells, YFP⁺CD4⁺CD44⁺CD62L⁻CD69⁻ T$_{em}$ cells, YFP⁺CD4⁺CD44⁺CD62L⁺ T$_{cm}$ cells, YFP⁻CD4⁺CD44⁺CD69⁺ T$_{rm}$ cells and YFP⁻CD4⁺CD44⁻ T$_{n}$ cells isolated from the kidneys of IL-17A$^{MI}$ mice (YFP⁺ T$_{m}$ cells: n = 6; YFP⁻ T$_{rm}$ cells: n = 4; T$_{n}$ cells: n = 3). **d,** Heatmap showing

the expression of ribosomal protein gene mRNA (analyzed with scRNA-seq data as in a). **e,** Protein-protein interaction enrichment analysis showing downregulated pathways in YFP⁺CD4⁺Sell⁻S1pr1⁻Klf2⁻ T$_{rm}$ cells (analyzed with scRNA-seq data as in a). **f,** Time course plots showing the frequency of IFN-γ⁺ and GM-CSF⁺ cells in CD4⁺CD44⁺CD69⁺ T$_{rm}$ cells, CD4⁺CD44⁺CD62L⁻CD69⁻ T$_{em}$ cells, CD4⁺CD44⁺CD62L⁺ T$_{cm}$ cells and CD4⁺CD44⁻ T$_{n}$ cells isolated from the kidneys of IL-17A$^{MI}$ mice (n = 6 for each group). Data are mean + s.e.m. Statistical analysis was performed using one-way ANOVA with Tukey's multiple comparison test; * $p < 0.05$, ** $p < 0.01$.

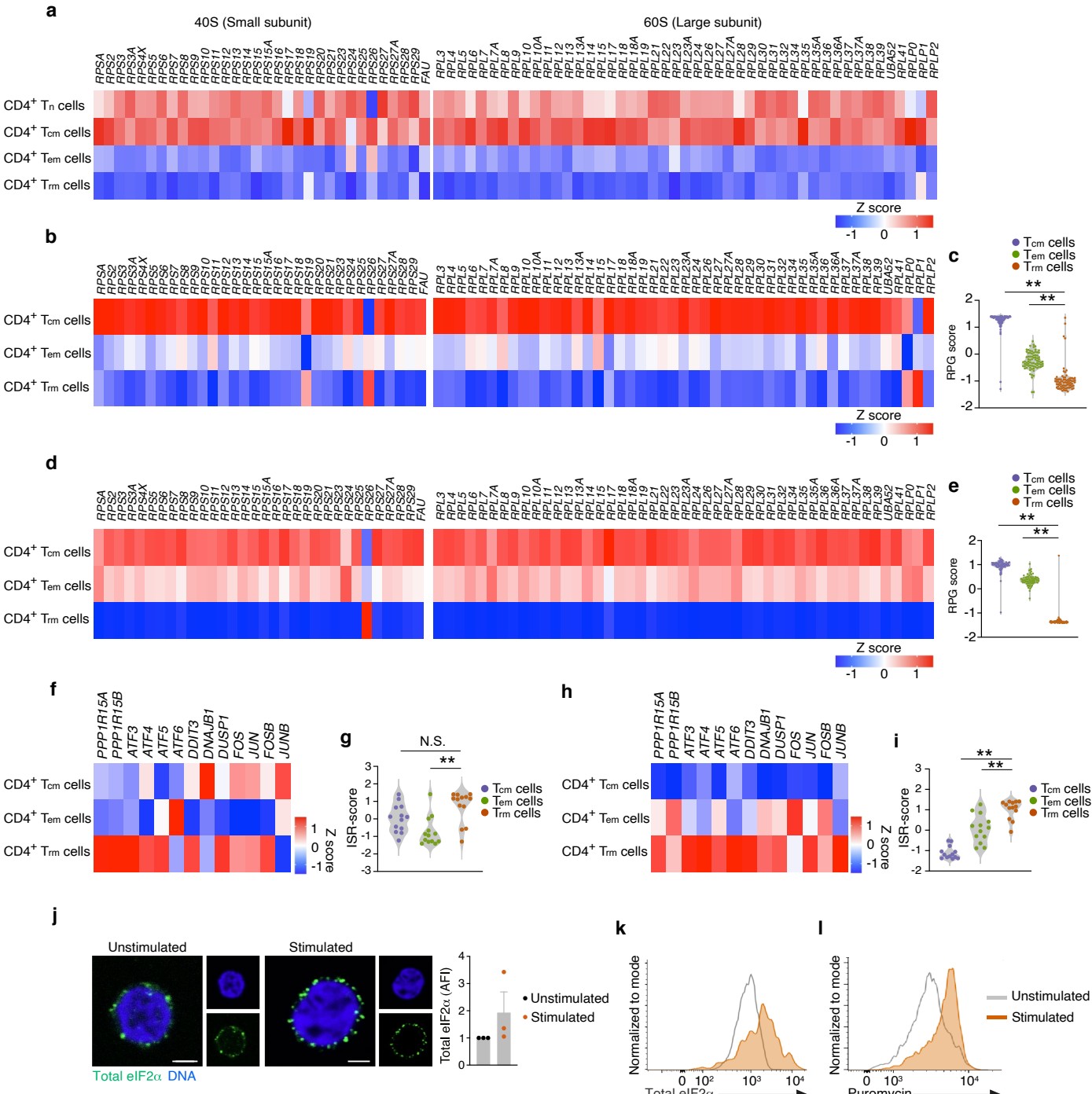

**Extended Data Fig. 5 | Ribosomal protein genes (RPGs) and integrated stress response (ISR)-associated genes in T cells. a,** Heatmap showing the expression of all RPGs in CD4[+] T$_{rm}$, T$_{em}$, T$_{cm}$ and T$_n$ cell clusters isolated from the human healthy kidney (scRNA-seq analysis of cohort 1). **b,c,** Heatmap showing the expression of all RPGs (**b**) and a violin plot showing RPG scores calculated based on the expression of all RPGs (**c**) in CD4[+] T$_{rm}$, T$_{em}$ and T$_{cm}$ cell clusters isolated from the human healthy colon. **d,e,** Heatmap showing the expression of all RPGs (**d**) and a violin plot showing RPG scores calculated based on the expression of all RPGs (**e**) in CD4[+] T$_{rm}$, T$_{em}$ and T$_{cm}$ cell clusters isolated from the human healthy lung. **f-i,** Heatmaps showing the expression of ISR-associated genes and violin plots showing the ISR scores calculated based on the ISR-associated genes in CD4[+] T$_{rm}$, T$_{em}$ and T$_{cm}$ cell clusters in the scRNA-seq data of human healthy colon

(**f,g**) and lung (**h,i**). **j,** Representative images of immunocytochemistry and a bar graph showing total eIF2α levels in sorted CD4[+]CD69[+] T$_{rm}$ cells isolated from the human healthy kidneys and unstimulated or stimulated with PMA+Iono (n = 3). Scale bars indicate 2 μm. **k,** Representative histogram showing total eIF2α levels in sorted CD4[+]CD69[+] T$_{rm}$ cells isolated from the human healthy kidneys and unstimulated or stimulated with CD3 + CD28 Ab for 2 days. **l,** Representative histogram showing global protein synthesis measured using a puromycin-uptake assay in sorted CD4[+]CD69[+] T$_{rm}$ cells isolated from the human healthy kidneys and unstimulated or stimulated with PMA+Iono for 2 hours. Data are mean + s.e.m. Statistical analysis was performed using one-way ANOVA with Tukey's multiple comparison test (c,e,g,i) or unpaired two-tailed t-test with Welch's correction (j); *$p < 0.05$, **$p < 0.01$.

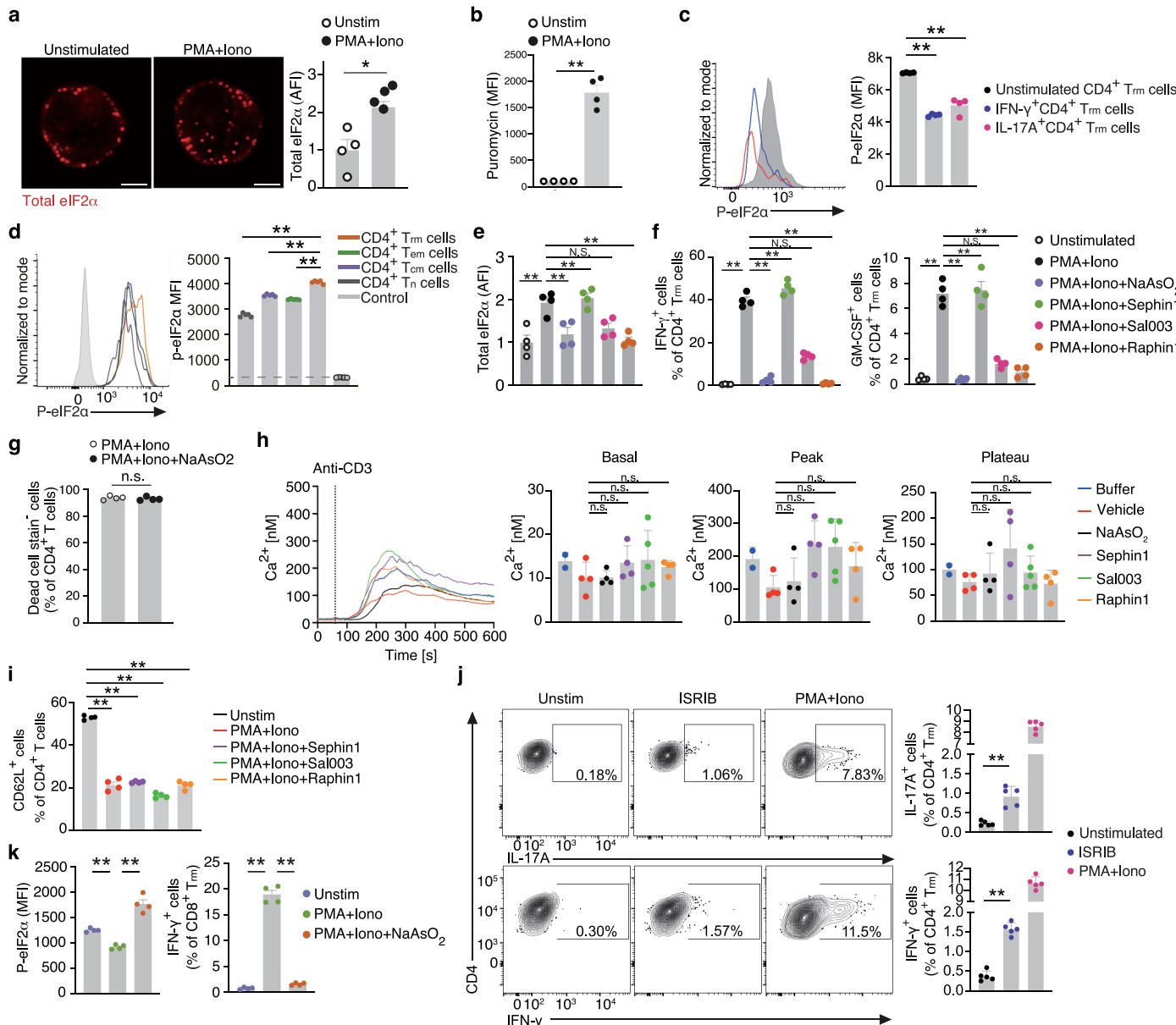

**Extended Data Fig. 6 | Analysis of the ISR in murine CD4⁺CD69⁺ T$_{rm}$ cells.**
**a**, Representative images of immunocytochemistry and a bar graph showing total eIF2α levels in CD4⁺CD69⁺ T$_{rm}$ cells isolated from the kidneys of WT^{MI} mice and unstimulated or stimulated with PMA+Iono for 3 hours (n = 4). Scale bars indicate 2 μm. **b**, Bar graph showing global protein synthesis in CD4⁺CD69⁺ T$_{rm}$ cells isolated from the kidney of WT^{MI} mice, unstimulated or stimulated with PMA+Iono for 3 hours and analyzed by a puromycin-uptake assay (n = 4). **c**, Representative histogram and quantification showing p-eIF2α levels in CD4⁺CD69⁺ T$_{rm}$ cells isolated from the kidneys of WT^{MI} mice and unstimulated or stimulated with PMA+Iono for 3 hours. p-eIF2α levels in IFN-γ⁺ T$_{rm}$ cells and IL-17A+ T$_{rm}$ cells were compared to those in unstimulated controls (n = 4). **d**, Representative histogram showing p-eIF2α levels in CD4⁺CD44⁺CD69⁺ T$_{rm}$ cells, CD4⁺CD44⁺CD62L⁻CD69⁻ T$_{em}$ cells, CD4⁺CD44⁺CD62L⁺ T$_{cm}$ cells and CD4⁺CD44⁻ T$_n$ cells isolated from the kidneys of WT^{MI} mice (n = 5). For the control group, MFI levels of anti-rabbit-647 Ab without primary Ab in CD4⁺CD44⁺ T$_m$ cells are shown. **e**, **f**, Bar graphs showing total eIF2α levels and the production of IFN-γ and GM-CSF in CD4⁺CD69⁺ T$_{rm}$ cells isolated from the kidneys of WT^{MI} mice and stimulated with PMA+Iono in the presence of vehicle, 500 μM NaAsO₂, 10 μM Sal003, 20 μM Sephin1 or 20 μM Raphin1 for 3 hours (n = 4). **g**, Bar graph showing

the frequency of dead stain- cells in CD4⁺CD69⁺ T$_{rm}$ cells isolated from the kidneys of WT^{MI} mice and stimulated with PMA+Iono in the presence of vehicle or NaAsO₂ for 3 hours (n = 4). **h**, Representative histogram and quantification showing intracellular Ca²⁺ concentration in CD4⁺ T cells isolated from the spleen of wild-type C57BL/6 mice, loaded with Fura2-AM and stimulated with anti-CD3 Ab in the presence of indicated compounds (n = 4 or 5). **i**, Bar graph showing the surface CD62L expression in CD4⁺ T cells isolated from the spleen of wild-type C57BL/6 mice, and unstimulated or stimulated with PMA+Iono in the presence of indicated compounds for 3 hours (n = 4). **j**, Representative flow cytometry plots showing the production of IL-17A and IFN-γ in CD4⁺CD69⁺ T$_{rm}$ cells isolated from the kidneys of WT^{MI} mice and cultured ex vivo in the presence of ISRIB for overnight or PMA+Iono for two hours (n = 5). **k**, Bar graphs showing p-eIF2α levels and IFN-γ production in CD8⁺CD69⁺ T$_{rm}$ cells isolated from the small intestine of wild-type C57BL/6 mice and unstimulated or stimulated with PMA+Iono in the presence or absence of 500 μM NaAsO₂ for 3 hours (n = 4). Data are mean + s.e.m. Statistical analysis was performed using one-way ANOVA with Tukey's multiple comparison test (c-f, h-k) or unpaired two-tailed t-test with Welch's correction (a,b,g); * $p < 0.05$, ** $p < 0.01$.

a

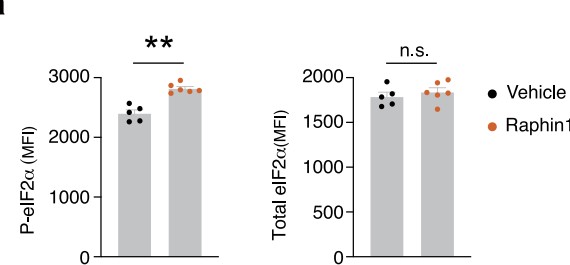

b

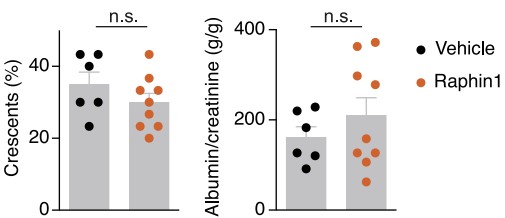

**Extended Data Fig. 7 | Raphin1 treatment in vivo. a**, Bar graphs showing phospho-eIF2α and total eIF2α levels in CD4$^+$CD69$^+$ T$_{rm}$ cells isolated from the kidneys of cGN$^{MI}$ mice. Kidney T$_{rm}$ cells were analyzed 4 hours after the last administration of vehicle (n = 5) or raphin1 (n = 6) by oral gavage. eIF2α levels were measured by flow cytometry. **b**, Bar graphs showing the quantification of glomerular crescent scores and urinary albumin to creatinine ratio in C57BL/6

mice. cGN was induced in wild-type C57BL/6 mice and vehicle (n = 6) or an eIF2α phosphatase inhibitor, raphin1 (n = 9), was given twice daily by oral gavage between d0 and d10 of cGN. Kidney function was evaluated at day 10 day after cGN induction. Data are mean + s.e.m. Statistical analysis was performed using unpaired two-tailed t-test with Welch's correction (a,b); *$p$ < 0.05, **$p$ < 0.01.

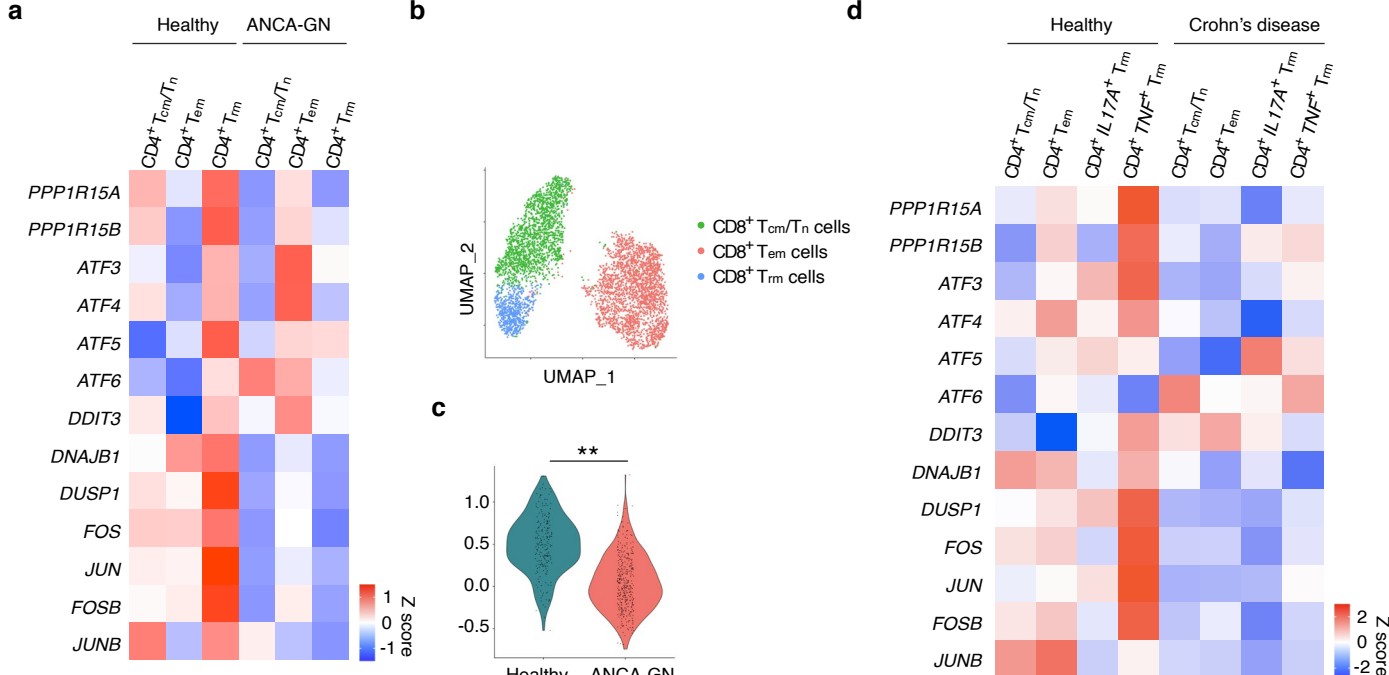

**Extended Data Fig. 8 | scRNA-seq analysis of T$_{rm}$ cells from patients with ANCA-GN and Crohn's disease. a**, Heatmap showing the expression of ISR-associated genes, *PPP1R15A, PPP1R15B, ATF3, ATF4, ATF5, ATF6, DDIT3, DNAJB1, DUSP1, FOS, JUN, FOSB* and *JUNB*, in *CD4$^+$KLF2$^-$*CD69$^+$ T$_{rm}$ cells, *CD4$^+$SELL$^-$KLF2$^+$* CD69$^-$ T$_{em}$ cells and CD4$^+$SELL$^+$KLF2$^+$CD69$^-$ T$_{cm}$/T$_n$ cells isolated from the kidney of healthy controls and ANCA-GN patients. **b**, UMAP plot showing *CD8A$^+$KLF2$^-$* CD69$^+$ T$_{rm}$ cells, *CD8A$^+$SELL$^-$KLF2$^+$*CD69$^-$ T$_{em}$ cells and *CD8A$^+$SELL$^+$KLF2$^+$* CD69$^-$ T$_{cm}$/T$_n$ cells isolated from the kidney of healthy controls and ANCA-GN

patients and analyzed with scCITE-seq. **c**, Violin plot showing the ISR score in *CD8A$^+$KLF2$^-$*CD69$^+$ T$_{rm}$ cells isolated from the kidney of healthy controls and ANCA-GN patients. **d**, Heatmap showing the expression of ISR-associated genes in *CD4$^+$S1PR1$^-$IL17A$^+$* T$_{rm}$ cells, *CD4$^+$S1PR1$^-$TNF$^+$* T$_{rm}$ cells, *CD4$^+$SELL$^-$S1PR1$^+$* T$_{em}$ cells and *CD4$^+$SELL$^+$CCR7$^+$S1PR1$^+$* T$_{cm}$/T$_n$ cells isolated from the colon of healthy controls and patients with Crohn's disease. Statistical analysis was performed using unpaired two-tailed t-test with Welch's correction; * *p* < 0.05, ** *p* < 0.01.

Christian Krebs
Hans-Willi Mittrücker

# Reporting Summary

## Statistics

For all statistical analyses, confirm that the following items are present in the figure legend, table legend, main text, or Methods section.

| n/a | Confirmed | |
|---|---|---|
| ☐ | ☒ | The exact sample size (*n*) for each experimental group/condition, given as a discrete number and unit of measurement |
| ☐ | ☒ | A statement on whether measurements were taken from distinct samples or whether the same sample was measured repeatedly |
| ☐ | ☒ | The statistical test(s) used AND whether they are one- or two-sided *Only common tests should be described solely by name; describe more complex techniques in the Methods section.* |
| ☒ | ☐ | A description of all covariates tested |
| ☐ | ☒ | A description of any assumptions or corrections, such as tests of normality and adjustment for multiple comparisons |
| ☐ | ☒ | A full description of the statistical parameters including central tendency (e.g. means) or other basic estimates (e.g. regression coefficient) AND variation (e.g. standard deviation) or associated estimates of uncertainty (e.g. confidence intervals) |
| ☐ | ☒ | For null hypothesis testing, the test statistic (e.g. *F*, *t*, *r*) with confidence intervals, effect sizes, degrees of freedom and *P* value noted *Give P values as exact values whenever suitable.* |
| ☒ | ☐ | For Bayesian analysis, information on the choice of priors and Markov chain Monte Carlo settings |
| ☒ | ☐ | For hierarchical and complex designs, identification of the appropriate level for tests and full reporting of outcomes |
| ☒ | ☐ | Estimates of effect sizes (e.g. Cohen's *d*, Pearson's *r*), indicating how they were calculated |

*Our web collection on statistics for biologists contains articles on many of the points above.*

## Software and code

Policy information about availability of computer code

| Data collection | FACS symphony<br>Zeiss LSM800 confocal microscope |
|---|---|
| Data analysis | FlowJo software 10.5.3<br>GraphPad Prism 8, 10<br>R studio, Jupyter Notebook<br>ZEN software (Carl Zeiss, Jena, Germany) |

For manuscripts utilizing custom algorithms or software that are central to the research but not yet described in published literature, software must be made available to editors and reviewers. We strongly encourage code deposition in a community repository (e.g. GitHub). See the Nature Portfolio guidelines for submitting code & software for further information.

## Data

Policy information about availability of data

All manuscripts must include a data availability statement. This statement should provide the following information, where applicable:

- Accession codes, unique identifiers, or web links for publicly available datasets
- A description of any restrictions on data availability
- For clinical datasets or third party data, please ensure that the statement adheres to our policy

scRNA-seq data of T cells from human healthy kidney tissue and Trm cell-induced murine kidneys are available at GSE270533 and FigShare webpage (https://figshare.com/s/7912de1afc7fd5bbefd4). The scRNA-seq data of the healthy colon and lung were obtained from GSE157477 and Cross-Tissue Immune Cell Atlas (https://www.tissueimmunecellatlas.org/#publication), respectively. The transcriptome data of European Renal cDNA Bank (kidney biopsy transcriptome datasets) is available from GSE104948 and refine.bio webpage. (https://www.refine.bio/experiments/GSE104948/glomerular-transcriptome-from-european-renal-cdna-bank-subjects-and-living-donors). The transcriptome datasets of colon tissues from Crohn's disease and healthy individuals are available from GSE109142. For the reference genome, refdata-cellranger-hg19-1.2.0 (human) and refdata-gex-mm10-2020-A (mouse) were used.All other data needed to verify the study's conclusions are contained in the manuscript or the Supplementary Materials.

## Research involving human participants, their data, or biological material

Policy information about studies with human participants or human data. See also policy information about sex, gender (identity/presentation), and sexual orientation and race, ethnicity and racism.

| | |
|---|---|
| Reporting on sex and gender | No sex- and gender-based analyses have been performed because of low sample sizes. |
| Reporting on race, ethnicity, or other socially relevant groupings | not applicable |
| Population characteristics | No data on population characteristics was collected. |
| Recruitment | Human kidney tissue was obtained from patients included in the Hamburg GN Registry, the European Renal cDNA Bank, and the Clinical Research Unit 228 (CRU 228) ANCA-GN cohort. |
| Ethics oversight | The research studies were conducted with the approval of the Ethik-Kommission der Ärztekammer Hamburg (the local ethics committee of the Chamber of Physicians in Hamburg) and in compliance with the ethical principles outlined in the Declaration of Helsinki. |

Note that full information on the approval of the study protocol must also be provided in the manuscript.

# Field-specific reporting

Please select the one below that is the best fit for your research. If you are not sure, read the appropriate sections before making your selection.

☒ Life sciences ☐ Behavioural & social sciences ☐ Ecological, evolutionary & environmental sciences

For a reference copy of the document with all sections, see nature.com/documents/nr-reporting-summary-flat.pdf

# Life sciences study design

All studies must disclose on these points even when the disclosure is negative.

| | |
|---|---|
| Sample size | Sample sizes were determined based on prior research conducted in our laboratories to use sufficient numbers of mice or cells in each group. |
| Data exclusions | no data were excluded |
| Replication | All findings were confirmed twice or more |
| Randomization | Mice were matched for age and sex before randomization, |
| Blinding | Investigators were aware of the group allocation because the treatment groups needed to be clear when performing the experiments. |

# Reporting for specific materials, systems and methods

We require information from authors about some types of materials, experimental systems and methods used in many studies. Here, indicate whether each material, system or method listed is relevant to your study. If you are not sure if a list item applies to your research, read the appropriate section before selecting a response.

## Materials & experimental systems

| n/a | Involved in the study |
|---|---|
| ☐ | ☒ Antibodies |
| ☐ | ☒ Eukaryotic cell lines |
| ☒ | ☐ Palaeontology and archaeology |
| ☐ | ☒ Animals and other organisms |
| ☒ | ☐ Clinical data |
| ☒ | ☐ Dual use research of concern |
| ☒ | ☐ Plants |

## Methods

| n/a | Involved in the study |
|---|---|
| ☒ | ☐ ChIP-seq |
| ☐ | ☒ Flow cytometry |
| ☒ | ☐ MRI-based neuroimaging |

## Antibodies

**Antibodies used**

human: CD45 (HI30, BioLegend), CD3 (OKT3, BioLegend), CD4 (RPA-T4, BioLegend), CD8 (RPA-T8, BioLegend), CD69 (FN50, BioLegend), CD45RA (HI100, BioLegend), GM-CSF (BVD2-21C11, BioLegend), IL-17A (BL168, BioLegend), IL-17F (SHLR17, eBioscience), IFN-γ (4S.B3, BioLegend), TNF-α (MAB11, BioLegend), CCR7/CD197 (Go43H7, BioLegend);
mouse: CD45 (30-F11, BioLegend), CD3 (145-2C11, BioLegend), CD4 (RM4-5, BioLegend), CD8 (53-6.7, BioLegend), GM-CSF (MP1-22E9, BioLegend), IL-17A (TC11-18H10, BioLegend), IL-17F (9D3.1C8, BioLegend), IL-22 (poly5164, BioLegend), IFN-γ (XMG1.2, BioLegend), TNF-α (MP6-XT22, BioLegend). All antibodies were diluted at a ratio of 1:100 to 1:200.

**Validation**

Antibodies used in this study are commercially available and have been validated by the manufacturers. Validation statements are provided on the manufacture's website.
human antibody CD45 HI30 BioLegend
https://www.biolegend.com/de-at/products/purified-anti-human-cd45-antibody-710
human antibody CD3 OKT3 BioLegend
https://www.biolegend.com/de-de/products/purified-anti-human-cd3-antibody-3642
human antibody CD4 RPA-T4 BioLegend
https://www.biolegend.com/de-at/products/purified-anti-human-cd4-antibody-830
human antibody CD8 RPA-T8 BioLegend
https://www.biolegend.com/de-de/products/purified-anti-human-cd8a-antibody-839
human antibody CD69 FN50 BioLegend
https://www.biolegend.com/en-ie/products/pe-anti-human-cd69-antibody-1672
human antibody CD45RA HI100 BioLegend
https://www.biolegend.com/nl-be/products/purified-anti-human-cd45ra-antibody-689
human antibody GM-CSF BVD2-21C11 BioLegend
https://www.biolegend.com/en-ie/products/pe-anti-human-gm-csf-antibody-916?GroupID=BLG1622
human antibody IL-17A BL168 BioLegend
https://www.biolegend.com/en-ie/products/purified-anti-human-il-17a-antibody-4442
human antibody IL-17F SHLR17 eBioscience
https://www.thermofisher.com/antibody/product/IL-17F-Antibody-clone-SHLR17-Monoclonal/16-7169-82
human antibody IFN-γ 4S.B3 BioLegend
https://www.biolegend.com/de-de/products/pe-anti-human-ifn-gamma-antibody-1011?GroupID=BLG10006
human antibody TNF-α MAB11 BioLegend
https://www.biolegend.com/ja-jp/products/purified-anti-human-tnf-alpha-antibody-1347
human antibody CCR7/CD197 Go43H7 BioLegend
https://www.biolegend.com/de-at/products/pe-anti-human-cd197-ccr7-antibody-7498
mouse antibody CD45 30-F11 BioLegend
https://www.biolegend.com/nl-be/products/purified-anti-mouse-cd45-antibody-102
mouse antibody CD3 145-2C11 BioLegend
https://www.biolegend.com/fr-lu/products/apc-anti-mouse-cd3epsilon-antibody-21
mouse antibody CD4 RM4-5 BioLegend
https://www.biolegend.com/nl-nl/products/purified-anti-mouse-cd4-antibody-484
mouse antibody CD8 53-6.7 BioLegend
https://www.biolegend.com/nl-be/products/apc-anti-mouse-cd8a-antibody-150
mouse antibody GM-CSF MP1-22E9 BioLegend
https://www.biolegend.com/ja-jp/products/pe-anti-mouse-gm-csf-antibody-958
mouse antibody IL-17A TC11-18H10 BioLegend
https://www.biolegend.com/nl-be/products/purified-anti-mouse-il-17a-antibody-1634
mouse antibody IL-17F 9D3.1C8 BioLegend
https://www.biolegend.com/fr-ch/products/pe-anti-mouse-il-17f-antibody-6964
mouse antibody IL-22 poly5164 BioLegend
https://www.biolegend.com/nl-be/products/pe-anti-mouse-il-22-antibody-6486
mouse antibody IFN-γ XMG1.2 BioLegend
https://www.biolegend.com/fr-ch/products/purified-anti-mouse-ifn-gamma-antibody-998
mouse antibody TNF-α MP6-XT22 BioLegend
https://www.biolegend.com/de-de/products/pe-anti-mouse-tnf-alpha-antibody-978?GroupID=GROUP24

# Eukaryotic cell lines

Policy information about cell lines and Sex and Gender in Research

| | |
|---|---|
| Cell line source(s) | HEK293T (https://www.atcc.org/products/crl-3216) |
| Authentication | not authenticated |
| Mycoplasma contamination | Cells which tested negative for mycoplasma were used for experiments |
| Commonly misidentified lines<br>(See ICLAC register) | No commonly misidentified cell lines were used. |

# Animals and other research organisms

Policy information about studies involving animals; ARRIVE guidelines recommended for reporting animal research, and Sex and Gender in Research

| | |
|---|---|
| Laboratory animals | Male mice on a C57BL/6 background, aged between 8 and 16 weeks, were used. Mice were maintained in specific pathogen-free conditions with controlled humidity and temperature with a light/dark cycle of 12h each. |
| Wild animals | not used |
| Reporting on sex | only males were used |
| Field-collected samples | No field-collected samples were used in this study. |
| Ethics oversight | Experiments with mice followed the national guidelines, and local ethics committees (Behörde für Justiz und Verbraucherschutz Hamburg) approved the research protocols. |

Note that full information on the approval of the study protocol must also be provided in the manuscript.

# Plants

| | |
|---|---|
| Seed stocks | n.a. |
| Novel plant genotypes | n.a. |
| Authentication | n.a. |

# Flow Cytometry

## Plots

Confirm that:

☒ The axis labels state the marker and fluorochrome used (e.g. CD4-FITC).

☒ The axis scales are clearly visible. Include numbers along axes only for bottom left plot of group (a 'group' is an analysis of identical markers).

☒ All plots are contour plots with outliers or pseudocolor plots.

☒ A numerical value for number of cells or percentage (with statistics) is provided.

## Methodology

| | |
|---|---|
| Sample preparation | Single-cell suspensions were obtained from kidney and blood samples to isolate and analyze human leukocytes. Kidney tissue was enzymatically digested with collagenase D at 0.4 mg/ml (Roche, Mannheim, Germany) and DNase I (10 µg/ml, Sigma-Aldrich, Saint Louis, MO) in RPMI 1640 medium at 37° C for 30 minutes, followed by dissociation with gentleMACS (Miltenyi Biotec). Blood samples were separated using Leucosep tubes (Greiner Bio-One, Kremsmünster, Austria). Samples were filtered through a 30-µm filter (Partec, Görlitz, Germany) prior to antibody staining and flow cytometry.<br>Cells from murine spleens were isolated by squashing the organ through a 70-µm cell strainer. Erythrocytes were lysed using a lysis buffer (155 mM NH4Cl, 10 mM KHCO3, 10 µM EDTA, pH 7.2). To isolate renal lymphocytes from mice, kidneys were enzymatically digested with 400 µg/ml collagenase D (Roche) and 10 U/ml DNase I (Sigma-Aldrich) at 37°C for 30 min. |

Subsequently, leukocytes were isolated by density gradient centrifugation using 37% Easycoll (Merck Millipore) and a filtration step using a 30-μm cell strainer (Partec). T cell isolation from the intestine is described previously. Briefly, murine intestine was cut longitudinally after removing Peyer's patches and adventitial fat. To collect intraepithelial lymphocytes, the intestine tissue was incubated in HBSS containing 1mM dithioerythritol followed by a dissociation step using 1 mM EDTA for 20 min at 37° C. To collect lamina propria lymphocytes, the tissue was cut into small pieces and incubated for 45 min at 37° C in HBSS supplied with 1 mg/ml collagenase and 10 U/ml DNase I. Leukocytes were further enriched by Percoll gradient. Intraepithelial cells and lamina propria lymphocytes were pooled for analysis.

Instrument | FACS symphony

Software | FlowJo

Cell population abundance | Cell populations were abundant enough for any of the analysis. Over 5000 cells of each target cell population were detected from the kidney and intestine of human and mice.

Gating strategy | The initial gate on FSC/SSC plots was set to remove cell debris and single cells were gated according to FSC-W and FSC-H. After gating on live cells, target cell populations were gated. Trm cells were gated as CD45+ CD4+ CD44+ CD69+ CD62L-; Tem cells as CD45+ CD4+ CD44+ CD69- CD62L-; Tcm cells as CD45+ CD4+ CD44+ CD69- CD62L+; Naïve T cells as CD45+ CD4+ CD44- CD69- CD62L+.

☒ Tick this box to confirm that a figure exemplifying the gating strategy is provided in the Supplementary Information.

