## [Peer Review File · Nature Immunology]

The integrated stress response pathway controls cytokine production in tissue-resident memory CD4+ T cells

Corresponding Author: Professor Ulf Panzer

Version 0:

Decision Letter:

15th Apr 2024

Dear Dr. Panzer,

Your Article, "The integrated stress response/eIF2 α pathway controls cytokine production in tissue-resident memory CD4+ T cells" has now been seen by 2 referees. While we find your work of considerable potential interest, the reviewers have raised concerns that must be addressed. As such, we cannot accept the current version of the manuscript for publication, but would be happy to consider a revised version that addresses these concerns, as long as novelty is not compromised in the interim.

Please revise the manuscript to address all issues raised by the referees. At resubmission, please include a point-by-point "Response to referees" detailing how you have addressed each referee comment (please specify the page and figure number where the new data can be found in the revised manuscript and please highlight the changes in the manuscript as well). This response will be sent back to the referees along with the revised manuscript.

In addition, please include a revised version of any required reporting checklist. It will be available to referees (and, potentially, statisticians) to aid in their evaluation if the manuscript goes back for peer review. A revised checklist is essential for re-review of the paper. The Reporting Summary can be found here: <https://www.nature.com/documents/nr-reporting-summary.pdf>

Link Redacted

We hope to receive a suitably revised manuscript within 4 months. If you cannot send it within this time, please let us know. We will be happy to consider your revision so long as nothing similar has been accepted for publication at Nature Immunology or published elsewhere.

Nature Immunology is committed to improving transparency in authorship. As part of our efforts in this direction, we are now requesting that all authors identified as 'corresponding author' on published papers create and link their Open Researcher and Contributor Identifier (ORCID) with their account on the Manuscript Tracking System (MTS), prior to acceptance. ORCID helps the scientific community achieve unambiguous attribution of all scholarly contributions. You can create and link your

ORCID from the home page of the MTS by clicking on 'Modify my Springer Nature account'. For more information please visit please visit www.springernature.com/orcid.

Thank you for the opportunity to review your work.

Sincerely,

Ioana Staicu, Ph.D.
Senior Editor
Nature Immunology

Tel: 212-726-9207
Fax: 212-696-9752
www.nature.com/ni

Reviewers' Comments:

Reviewer #1:

Remarks to the Author:

This manuscript by Asada et al documents the presence of untranslated mRNA for cytokines in mouse and human tissue resident memory T cells. Its presents a series of experiments which support the hypothesis that translation initiation is repressed in TRM via the inactivation of the EIF2 complex. This is a well-studied and understood mechanism mediated by phosphorylation of EIF2a as part of the integrated stress response. It is novel and of interest that TRM appear to operate an ISR. Results in disease models and patients' datasets implicate the ISR in autoimmune/inflammatory kidney disease. This is a clearly written manuscript which tackles a neglected and very complex molecular process linking immune homeostasis, memory responses and immunopathology.

The hypothesis that in steady-state conditions, cytokine mRNA translation in TRM cells is suppressed by the ISR pathway can be supported strongly by testing the effects of small molecules that bypass Eif2a-P (drugs such as ISRIB, DNL343). Do these promote cytokine production in TRMs? Remarkably, given that drugs of this kind are being used in preclinical models there seems to be little if any information on how they affect cytokine production. This would be of broad interest.

Providing more evidence to confirm that eIF2 α dephosphorylation specifically affects the translation of preformed cytokine mRNA. This could include looking further at the timings of dephosphorylation (does dephosphorylation occur prior to cytokine production?), showing that the dephosphorylation inhibitors are not altering the activation state of the T cells (do these drugs inhibit T cell activation?), and by showing that Trm (or indeed any CD4+ T cell) that is producing cytokines has reduced eIF2 α phosphorylation.

Please clarify the interpretation of supplementary figure 6 where it is shown that there is not much of a difference in phospho-eIF2 between the memory subsets. Doesn't this suggest the pathway is not preferentially active in Trm? It is apparent that the abundance of cytokine mRNA differs between these subsets> is that likely to contribute to the difference in cytokine production?

When the authors write about cytokine scores (Figure 2 and 5) it is not fully clear which cytokines are considered. Is it a specific subset or all cytokines? Do the other memory subsets show any poised mRNAs for example interferon gamma or Ccl5 which has been reported to be present in unactivated memory T cells?

In figure 5 comparing healthy and patient samples the ISR score seems to be reduced in patient samples which is consistent with the authors model for the steady state in Trm. Is the absence of an increased Cytokine score shown for Crohn's disease or ANCA GN consistent with the findings in figure 1. Is this result expected?

Minor Points

It is worth considering the extent to which reduced ribosomal biogenesis contributes to repression of cytokine production in TRM and what this says about the activity of pathways such as MTOR. The link with the TOR pathway may be less elusive than suggested in the discussion as its repression prevents eukaryotic initiation factor 4F formation.

Care should be taken when extending conclusions from IL17a to all cytokines (p12) in the absence of supporting data.

What are the numbers on the scalebar in Fig 1c?

Is the patient data distinguishing T cell subsets or only analyzing bulk T cells?

In Figure 4 a-c it is not clear what n refers to. Is it number of cells, or experiments, or images?

Figure 2h Statistics are missing for TCM vs TTM comparison.

Reviewer #2:

Remarks to the Author:

The manuscript "The integrated stress response /eIF2 α pathway controls cytokine production in tissue-resident memory CD4+ T cells" by Asada et al, provides mechanistic insights into the alertness and rapid responses in resident memory T (Trm) cells that are lodged in homeostatic tissues. Through analysis of descriptive scRNASeq data combined with elegant experiments in murine model of glomeronephritis, the authors highlight that pre-formed mRNA for several cytokines are stored in Trm cells in human kidneys, colon and lung. In a series of invitro and in vivo experiments, the authors show a role for integrated stress response (ISR)/ eukaryotic initiation factor 2 α (eIF2 α) pathway in regulating transcription of type 1 and type 3 cytokine production IF2 α , thereby providing a rapid mechanism of cytokine secretion following TCR ligation. This study is timely and provides novel understanding on how Trm cells maintain alert in homeostatic tissues.

Comments:

In single cell CITE-seq data sets, Trm cells are mostly annotated by the low expression of SELL, KLF2 and S1PR1. Authors can consider scoring T cells using the pan-tissues CD4 Trm signature from BV Kumar et al, 2017 to validate the Trm annotation.

From Fig 1f,1i, it seems like a small proportion of Trm cells have measurable cytokine scope in human kidney and colon whereas the data from lung Trm indicates that the cytokine score is up in the whole population. A similar pattern is shown from murine kidney in Fig2c. Is this reflecting a higher alertness in lung-poised Trm cells? Are there subsets of Trm cells in the kidney (and or colon) that are more poised to storing cytokine mRNA in homeostatic conditions?

Different subsets of skin Trms preferentially secrete type 1 and type 3 cytokines (S Naik et al Nature 2015, Cheuk, et al., 2017), potentially governed by epigenetic wiring (SL Park, et al., 2023). In s Fig 8b, ISR genes expression differs in TNF+ Trm and IL17A+ Trm in human T cells from diseased tissues. Thus, it would be interesting to compare the ISR score between Trm cell subsets and TNF+ Trm and IL17A+ Trm in Fig 5.

The authors found that CD8 Trm cells were poised to type 1 interferons and based the manuscript on the more multipotent CD4 Trm cell population in kidney. Regardless, it would be interesting to know if the same mechanisms is regulating cytokine translation in CD8 T cells, potentially by invitro experiments mirroring the data in Figure 4 a-f as well as providing analyses of the CD8 compartment in figure 5.

Could the authors expand on the therapeutic strategy in figure 4i-n? From the figure legend, the glomerunephritis is induced after 2 months (60 days) and treatment given day 70-80, ie 10 days after onset of disease. Is this a model of chronic glomerulnephritis? Would it be possible to revert pathology by initiating treatment in a chronic inflammatory stage of the model? Following cessation of therapy, would the inflammation and pathology relapse?

Minor coments

Inconsistent CCR7 CITE-seq protein and gene expression in Fig1E heatmaps. Authors may consider checking the positive calling of CCR7+ cells compared to background based on raw CITE-seq protein counts.

Clarify type of data shown in the figure legend of Fig 3B.

Same colour code to be used for Tcm, Tem and Trm in Fig 2 and Supplementary Fig 4 for better clarification.

CITE-seq protein name of CD62L should be used instead of gene name, SELL in Fig 1, Supplementary Fig S1 and Fig 5. CITE-seq proteins, CD44 and CCR7 expressions are not shown in supplementary Fig 1C.

IL17A, IL17F, IFNG or TNF gene expressions are not shown in Fig 5b.

Type 1 and 3 cytokine responsive genes score shown for ANCA-GN in Fig 5F is not shown in the context of Crohn's disease.

Version 1:

Decision Letter:

19th Sep 2024

Dear Dr. Panzer,

Thank you for your response to the referees comments on your Article "The integrated stress response/eIF2 α pathway controls cytokine production in tissue-resident memory CD4+ T cells" here at Nature Immunology. Although we are interested in the possibility of publishing your study in Nature Immunology, the issues raised by the referees need to be addressed.

Please revise along the lines specified in your letter. Please include Trm cells from the kidney in all the required experiments, as indicated, and please perform at least three replicates for all experiments, including the detection of p-eIF2 α . In addition, we consider it is essential to increase the novelty of the manuscript beyond previous observations that this type of regulation of the mRNA pool has been described in memory T cells. This could be achieved by assessing the regulatory process described here in all memory T cell subsets and in Trm cell across multiple tissues, by including the

characterization of multiple cytokines and chemokines and by addressing experimentally the issues raised by referee #1 regarding the specificity of this process compared to translation of other mRNAs, redundancy with other translational control mechanisms and the effect of the size of the untranslated cytokine mRNA pool. As suggested, please revise to accurately describe your results and their implications in the manuscript text. At resubmission, please include a "Response to referees" detailing, point-by-point, how you addressed each referee comment. If no action was taken to address a point, you must provide a compelling argument. This response will be sent back to the referees along with the revised manuscript.

Please include a revised version of any required reporting checklist. It will be available to referees to aid in their evaluation. Reporting summary can be found here:
<https://www.nature.com/documents/nr-reporting-summary.pdf>

Please note, Extended Data figures and tables are online-only (appearing in the online PDF and full-text HTML version of the paper), peer-reviewed display items that provide essential background to the Article but are not included in the printed version of the paper due to space constraints or being of interest only to a few specialists. A maximum of ten Extended Data display items (figures and tables) is typically permitted. When re-submitting your manuscript, please ensure that any supplementary figures and tables that are more critical to the manuscript's conclusions are converted to Extended data to increase these data's visibility.

Link Redacted

We hope to receive your revised manuscript within 8-10 weeks. If you cannot send it within this time, please let us know. We will be happy to consider your revision so long as nothing similar has been accepted for publication at Nature Immunology or published elsewhere.

Nature Immunology is committed to improving transparency in authorship. As part of our efforts in this direction, we are now requesting that all authors identified as 'corresponding author' on published papers create and link their Open Researcher and Contributor Identifier (ORCID) with their account on the Manuscript Tracking System (MTS), prior to acceptance. ORCID helps the scientific community achieve unambiguous attribution of all scholarly contributions. You can create and link your ORCID from the home page of the MTS by clicking on 'Modify my Springer Nature account'. For more information please visit www.springernature.com/orcid.

Sincerely,

Ioana Staicu, Ph.D.
Senior Editor
Nature Immunology

Tel: 212-726-9207
Fax: 212-696-9752
www.nature.com/ni

Reviewers' Comments:

Reviewer #1 (Remarks to the Author):

I appreciate the authors experiments with ISRIB which appear to show that inhibiting the ISR promotes IFN γ and IL-17A protein production. I agree with the authors that this appears to be a “relatively weak effect”. For clarity of interpretation, I would expect to see a representative FACS plot as the 1% population could either be a very small change in MFI e.g. a “shoulder of a histogram” or a clearly defined population and seeing this essential for interpretation. Furthermore, with such a small change I would expect to see a positive control (e.g., the same cells activated with PMA/ionomycin) to clearly show what range is expected. This is even more important as within this manuscript as TRMs are not always isolated from the same tissue and there are clear differences in response between different experiments.

Overall, I am wondering about the differences in some experiments with regards to the frequency of cytokine production in Trms. This makes evaluation of the effect size of some of the pharmacological experiments challenging. It would be helpful if the ISRIB experiments would be presented in a way that one could relate the effect size of the usage of ISRIB on cytokine production to for example Trm activation. What is the proportional effect of ISR on translational suppression? If it is only minor as the authors suggest the claims in the introduction and abstract need to be readjusted and discussed critically. Why were the experiments done with Trm from small intestine and not from kidney as the other inhibitor experiments?

Given these concerns, I suggest the manuscript text and abstract be revised to more clearly reflect the data and the limits of its interpretation. Statements in the text especially in the abstract which states “In steady-state conditions, cytokine mRNA translation in Trm cells is suppressed by the integrated stress response (ISR)/ eukaryotic initiation factor 2 α (eIF2 α) pathway” and “These findings suggest that the ISR/eIF2 α pathway tightly controls effector cytokine production in Trm cells, identifying a hitherto unknown mechanism of Trm-cell regulation under homeostatic and inflammatory conditions.”

In their response to the review, they write “The relatively weak effect might be due to other translational regulation mechanisms still being active (e.g., mTOR pathway, RNA-binding proteins, and miRNAs)”. I suggest this needs to be clarified in the manuscript.

The manuscript discussion would benefit from discussion on how this work fits with prior studies in other non TRM memory T cell types. The regulation of translation of CCL5 (<https://pubmed.ncbi.nlm.nih.gov/12433367/> and <https://pubmed.ncbi.nlm.nih.gov/16983723/> and IFN γ (ref 22) has been shown for other memory subsets thus this mode of regulation of cytokine and chemokine transcripts is likely not a TRM specific process.

I note their comment “Because this manuscript emphasizes cytokine mRNA and does not investigate chemokines”. Is there a biological reason for only focusing on cytokine mRNA?

For the effects of inhibitor on activation they have “focused on signaling mechanisms that are independent from transcription and translation” but why not provide insight into whether some proteins are able to be newly formed?

Regarding the differences p-eIF2 α expression between memory populations, I am not convinced that the new graph indicates that Trms have clearly more p-eIF2 α .

They didn't respond to the original question about cytokine mRNA “It is apparent that the abundance of cytokine mRNA differs between these subsets> is that likely to contribute to the difference in cytokine production?”.

Further points for clarification

Please clarify what is the “control “ in figure S6d? Which cell type?

From what I can tell the representative FACS plot for part C (also in extended fig 6c) they have switch to using a linear scale for phospho-Eif2a, given that linear scales are very unusual I would expect this to made clearer.

In figure 2i,j it is not clear whether the Trms from renal and small intestine tissue from the different mouse models were stimulated with the same conditions since the frequencies of Il17 producing cells differ significantly.

Was the origin of Trm cells on which Sephin and Raphin used the same (Small intestine vs renal tissues)? This is not clearly explained in the figure legends and text. inhibitors were used on cells from small intestine but what about the other experiments in figure 4?

Overall, this is an import study, but there remain significant question marks with regards to specificity (TRMs vs non TRMs, cytokine mRNA vs overall translation) and redundancy with other translational control mechanisms. The conclusions drawn in the abstract are too strong based on what is shown, and the authors acknowledge this in the response to the reviews, but I do not find them to be sufficiently considered in the discussion.

Reviewer #2 (Remarks to the Author):

The authors have taken all my comments into account in the revised version.

Version 2:

Decision Letter:

Our ref: NI-A37595B

13th Dec 2024

Dear Dr. Panzer,

Thank you for submitting your revised manuscript "The integrated stress response/eIF2 α pathway controls cytokine production in tissue-resident memory CD4⁺ T cells" (NI-A37595B). It has now been seen by the original referee and their comments are below. We are happy to inform you that if you revise your manuscript appropriately according to our editorial requirements, your manuscript should be publishable in Nature Immunology. The editors consider the manuscript is better suited as a Letter and we'll prepare it for publication as such.

I will now pre-edit the current version of your paper. We will also perform detailed checks on your paper and will send you a checklist detailing our editorial and formatting requirements in about two weeks. Please do not upload the final materials and make any revisions until you receive this additional information from us.

If you had not uploaded a Word file for the current version of the manuscript, we will need one before beginning the editing process; please email that to immunology@us.nature.com at your earliest convenience.

In the meantime, please deposit all omic and code data into public repositories so that the accession codes are readily available to be added in the revised manuscript. We cannot accept the paper without the codes. In addition, the ORCID of ALL corresponding authors needs to be linked to their Nature account (this frequently causes delays at acceptance). Should you have any query or comments about ORCID, please do not hesitate to contact our editorial assistant at immunology@us.nature.com.

Thank you again for your interest in Nature Immunology. Please do not hesitate to contact me if you have any questions.

Sincerely,

Ioana Staicu, Ph.D.
Senior Editor
Nature Immunology

Tel: 212-726-9207
Fax: 212-696-9752
www.nature.com/ni

Reviewer #1 (Remarks to the Author):

The authors revisions are appreciated, and, in the discussion, it is more clearly considered in the final paragraph that there are other mechanisms in addition to the ISR which are part of the process that inhibits translation of cytokine mRNAs in memory T cells. The demonstration of translational repression of cytokines specifically in Trm are novel and the implications for human pathologies are intriguing. It remains unclear if or how the ISR is specifically targeting cytokines in memory T cells and the extent to which this inhibition can be separated from more general repression of translation by the ISR. This mechanistic understanding is probably beyond the scope of this manuscript and will require more investigation in the future.

Reviewer #1:

Remarks to the Author:

This manuscript by Asada et al documents the presence of untranslated mRNA for cytokines in mouse and human tissue resident memory T cells. Its presents a series of experiments which support the hypothesis that translation initiation is repressed in TRM via the inactivation of the EIF2 complex. This is a well-studied and understood mechanism mediated by phosphorylation of EIF2a as part of the integrated stress response. It is novel and of interest that TRM appear to operate an ISR. Results in disease models and patients' datasets implicate the ISR in autoimmune/inflammatory kidney disease. This is a clearly written manuscript which tackles a neglected and very complex molecular process linking immune homeostasis, memory responses and immunopathology.

We are thankful to the reviewer for the constructive and insightful comments.

Comments:

The hypothesis that in steady-state conditions, cytokine mRNA translation in TRM cells is suppressed by the ISR pathway can be supported strongly by testing the effects of small molecules that bypass Eif2a-P (drugs such as ISRIB, DNL343). Do these promote cytokine production in TRMs? Remarkably, given that drugs of this kind are being used in preclinical models there seems to be little if any information on how they affect cytokine production. This would be of broad interest.

ISRIB and DNL343, chemical compounds that bypass the effect of a strong bond between p-eIF2α and eIF2B, are known to accelerate mRNA translation during the integrated stress response. Following the reviewer's suggestion, we treated Trm cells with ISRIB and examined their cytokine production.

In *in vitro*-cultured CD4⁺ Trm cells, treatment with ISRIB slightly but significantly increased the production of IFN-γ and IL-17A. The relatively weak effect might be due to other translational regulation mechanisms still being active (e.g., mTOR pathway, RNA-binding proteins, and miRNAs). Yet, given this result, using ISRIB in a preclinical model or patients might slightly augment inflammation. This point needs to be addressed and carefully monitored if ISRIB is used in clinical trials. We have complemented the new data in revised Extended Data Fig. 6j and commented on this point in the discussion.

ISRIB induces cytokine production from Trm cells. CD4⁺ Trm cells isolated from the small intestine were treated with 500 nM ISRIB *in vitro*, and analyzed with flow cytometry for the production of IFN-γ and IL-17A.

Providing more evidence to confirm that eIF2 α dephosphorylation specifically affects the translation of preformed cytokine mRNA. This could include looking further at the timings of dephosphorylation (does dephosphorylation occur prior to cytokine production?), showing that the dephosphorylation inhibitors are not altering the activation state of the T cells (do these drugs inhibit T cell activation?), and by showing that Trm (or indeed any CD4⁺ T cell) that is producing cytokines has reduced eIF2 α phosphorylation.

We sincerely appreciate the valuable comments and suggestions. We performed the following experiments to verify that eIF2 α dephosphorylation regulates the translation of preformed cytokine mRNA in Trm cells.

A: "Timing": Dephosphorylation starts to occur before cytokine production.

First, we analyzed the phosphorylation of eIF2 α at different time points to examine the timing of eIF2 α dephosphorylation and cytokine production. Indeed, eIF2 α dephosphorylation occurred as early as 30 minutes after begin of Trm cell stimulation, while cytokine production increased 60 minutes of stimulation. We have added the new data in revised Fig. 4d.

eIF2 α dephosphorylation takes place before cytokine production in Trm cells. CD4⁺ Trm isolated from the small intestine were stimulated with PMA/ionomycin and analyzed for p-eIF2 α levels as well as cytokine production at different time points of stimulation.

B: T-cell "activation": Dephosphorylation inhibitors do not interfere with T-cell activation.

Next, we analyzed whether the dephosphorylation inhibitors impede activation of T cells. We focused on signaling mechanisms that are independent from transcription and translation. A critical step during T-cell activation is the increase in intracellular calcium. The dephosphorylation inhibitors (Sephin1, Sal003, and Raphin1) did not suppress the increase in calcium signaling after TCR stimulation. This data is shown in revised Extended Data Fig. 6h.

ISR-inducing compounds do not affect intracellular Ca²⁺ signals in primary CD4⁺ T cells upon TCR stimulation

Murine CD4⁺ T cells isolated from the spleen were loaded with Fura2-AM and incubated in Ca²⁺ buffer with either NaAsO₂ (500 μ M), Sephin1 (20 μ M), Sal003 (10 μ M), Raphin1 (20 μ M) or Ca²⁺ buffer and Vehicle as control 5 min prior to acquisition. Left: Mean intracellular Ca²⁺ traces upon anti-CD3 addition (10 μ g/mL) after 1 min. (Right) Analysis of the mean basal (at 30s), peak (at 260 s), and plateau (at 520 s) Ca²⁺ concentration.

In addition, we measured CD62L surface expression following T-cell activation. CD62L is expressed by resting T cells but is enzymatically shed from the cell surface upon T-cell activation (PMID: 12055263, 9680363). This process is mediated by activation of the protease ADAM17 and does not require mRNA transcription or translation (PMID: 28877252). The dephosphorylation inhibitors did not suppress shedding of CD62L after T-cell activation. Thus, we conclude that these inhibitors do not generally inhibit T-cell activation. This data is shown in revised Extended Data Fig. 6i.

eIF2 α phosphatase inhibitors do not affect activation induced CD62L shedding from primary CD4⁺ T cells. Murine CD4⁺ T cells isolated from the spleen were stimulated with PMA/ionomycin for two hours in the presence of eIF2 α phosphatase inhibitors, and analyzed with flow cytometry.

C: Moreover, in Trm cells stimulated with PMA/ionomycin for two hours, cytokine production was mainly observed in cells with low phospho-eIF2 α levels. These results further support the concept that eIF2 α dephosphorylation is required for effective execution of effector functions following T-cell activation. These data are presented in revised Extended Data Fig. 6d.

Cytokine production is observed mainly in Trm cells with low levels of phospho-eIF2 α . CD4⁺ Trm cells isolated from the small intestine were stimulated with PMA/ionomycin for two hours and analyzed with flow cytometry for phospho-eIF2 α levels and cytokine production. Left: representative plots for cytokine and phospho-eIF2 α staining of CD4⁺ Trm cells; right: % cytokine positive cells in p-eIF2 α low and high CD4⁺ Trm cells.

Please clarify the interpretation of supplementary figure 6 where it is shown that there is not much of a difference in phospho-eIF2α between the memory subsets. Doesn't this suggest the pathway is not preferentially active in Trm? It is apparent that the abundance of cytokine mRNA differs between these subsets> is that likely to contribute to the difference in cytokine production?

As pointed out by the reviewer, the difference in phospho-eIF2α levels between memory T-cell subsets was not very large in our flow cytometry analyses. For better interpretation, we repeated the measurement of p-eIF2α in memory T cells and included a staining control (secondary Ab-only). In the revised graph, both p-eIF2α levels and negative control levels are shown. When considering the control signals, the difference in p-eIF2α signals between different memory T cells becomes more evident. These data are presented in revised Extended Data Fig. 6c.

CD4⁺ Trm cells show higher levels of phospho-eIF2α compared to other memory T cell subsets. CD4⁺ T cells isolated from the kidney were analyzed with flow cytometry for p-eIF2α levels. Left: Overlay of representative histograms for p-eIF2α staining of different memory T-cell subsets; Right: Mean fluorescence intensity of p-eIF2α staining for T cell subsets. The dotted line indicates staining intensity of secondary Ab only.

When the authors write about cytokine scores (Figure 2 and 5) it is not fully clear which cytokines are considered. Is it a specific subset or all cytokines? Do the other memory subsets show any poised mRNAs for example interferon gamma or Ccl5 which has been reported to be present in unactivated memory T cells?

We thank the reviewer for this comment and apologize for the unclear description of the cytokine scores. The score included *IFNG*, *TNF*, *IL2*, *CSF2*, *IL17A*, *IL17F*, *IL22*, and *LTA* in Figs. 1, 2, and 5. This information has been included in the revised figure legends of Figs. 1, 2, and 5.

For *IFNG* and *CCL5*, we performed additional analyses and confirmed that these transcripts were expressed. Because this manuscript emphasizes cytokine mRNA and does not investigate chemokines, only UMAPs showing the cytokine mRNA expression are included in Extended Data Fig. 1 (*IFNG* has been shown in Extended Data Fig. 1c with other cytokine genes; *CCL5* is only shown for the reviewers).

CD4⁺ Trm cells express high levels of CCL5. UMAP plots showing the annotation of T cell clusters, *IFNG* expression, and *CCL5* expression.

In figure 5 comparing healthy and patient samples the ISR score seems to be reduced in patient samples which is consistent with the authors model for the steady state in Trm. Is the absence of an increased Cytokine score shown for Crohn's disease or ANCA GN consistent with the findings in figure 1. Is this result expected?

We are grateful for the reviewer's insightful observations and questions. As the reviewer noted, the decrease in the ISR score in patients with inflammatory diseases aligns with our observations that the ISR pathway facilitates the storage of cytokine mRNA. Although anticipated, this correlation remains an exciting aspect of our findings.

In contrast, the similar cytokine mRNA expression levels observed in both homeostatic and inflammatory conditions were unexpected. This observation could be explained by the baseline expression of cytokine mRNA in Trm cells, which has been previously shown under homeostatic conditions. Supporting this, Hombrink et al. (PMID: 27776108) reported that cytokine mRNA levels in Trm cells were comparable to those in blood Tem cells stimulated with anti-CD3/CD28 Abs. We therefore assume that the mRNA levels of cytokines in Trm cells are not substantially further up-regulated under inflammatory conditions. This point is now mentioned in the discussion of the revised manuscript (please see page 17).

Minor Points

It is worth considering the extent to which reduced ribosomal biogenesis contributes to repression of cytokine production in TRM and what this says about the activity of pathways such as MTOR. The link with the TOR pathway may be less elusive than suggested in the discussion as its repression prevents eukaryotic initiation factor 4F formation.

We thank the reviewer for the insightful comments. As pointed out, resting Trm cells show lower expression of ribosomal protein genes and downregulation of ribosome biogenesis pathways. This indicates that Trm cells may be less capable of producing proteins under unstimulated conditions. We speculate that Trm cells upregulate ribosome biogenesis upon activation, as previously reported in other T-cell populations (PMID: 36598533). Although the regulation of the ribosome biogenesis pathway in Trm cells (ISR, mTOR, etc.) is of great interest, it is beyond the scope of our present manuscript.

The mTOR pathway, as commented, is involved in eIF4 complex formation and regulation of mRNA translation. Therefore, although it uses regulatory mechanisms that differ from those of the ISR, it regulates global protein synthesis similarly to the ISR. We assume there could be overlapping or redundant roles between ISR and mTOR in Trm cells. Converging regulation by mTORC1 and eIF2 α (ISR and mTOR share translation targets) has been reported in the field of cell biology (Klann et al. Functional Translatome Proteomics Reveal Converging and Dose-Dependent Regulation by mTORC1 and eIF2 α . Mol Cell. 2020). The role of mTOR in Trm cells has yet to be addressed.

Care should be taken when extending conclusions from IL17a to all cytokines (p12) in the absence of supporting data.

We agree with the reviewer that caution must be exercised in extrapolating the results from IL-17A to other cytokines. The sentences have been modified accordingly in the revised text on page 13.

What are the numbers on the scalebar in Fig 1c?

The scale bar in Figure 1c reflects the Z score of gene expression. This information has now been included in revised figures.

Is the patient data distinguishing T cell subsets or only analyzing bulk T cells?

With the exception of Fig. 5f,k, and Extended Data Fig. 3c,d, all mRNA expression analyses were performed based on single cell sequencing data (scRNA-seq), enabling us to distinguish different T-cell subsets.

In Figure 4 a-c it is not clear what n refers to. Is it number of cells, or experiments, or images?

We apologize for not presenting enough information. n refers to the number of experiments. We included this information in the revised figure legend.

Figure 2h Statistics are missing for TCM vs TTM comparison.

The statistical information on comparisons between Tcm and Ttm is now included in revised Fig. 2h.

Reviewer: 2

Remarks to the Author:

The manuscript “The integrated stress response /eIF2a pathway controls cytokine production in tissue-resident memory CD4+ T cells” by Asada et al, provides mechanistic insights into the alertness and rapid responses in resident memory T (Trm) cells that are lodged in homeostatic tissues. Through analysis of descriptive scRNASeq data combined with elegant experiments in murine model of glomeronephritis, the authors highlight that pre-formed mRNA for several cytokines are stored in Trm cells in human kidneys, colon and lung. In a series of invitro and in vivo experiments, the authors show a role for integrated stress response (ISR)/ eukaryotic initiation factor 2α (eIF2α) pathway in regulating transcription of type 1 and type 3 cytokine production IF2a, thereby providing a rapid mechanism of cytokine secretion following TCR ligation. This study is timely and provides novel understanding on how Trm cells maintain alert in homeostatic tissues.

We thank the reviewer for the positive and helpful comments.

In single cell CITE-seq data sets, Trm cells are mostly annotated by the low expression of SELL, KLF2 and S1PR1. Authors can consider scoring T cells using the pan-tissues CD4 Trm signature from BV Kumar et al, 2017 to validate the Trm annotation.

Following the reviewer’s suggestion, we implemented the CD4⁺ Trm signature score in single-cell CITE-seq datasets based on the manuscript by Kumar et al. We confirmed that the CD4⁺ T cell cluster annotated as Trm cells in our annotation had significantly higher Trm signature scores than other subsets, corroborating our Trm cell annotation. As both our annotation method and the Trm signature score by Krumar et al. identify the same cluster as Trm cells, our conclusion did not change. These data are shown only for the reviewers.

Trm signature from Kumar et al. is high in CD4 Trm cluster. (Left) Violin plot showing the Trm signature calculated based on Kumar et al., 2017. (Right) Heatmap showing the Trm signature and non-Trm signature genes based on Kumar et al., 2017.

From Fig 1f,1i, it seems like a small proportion of Trm cells have measurable cytokine scope in human kidney and colon whereas the data from lung Trm indicates that the cytokine score is up in the whole population. A similar pattern is shown from murine kidney in Fig2c. Is this reflecting a higher alertness in lung-poised Trm cells? Are there subsets of Trm cells in the kidney (and or colon) that are more poised to storing cytokine mRNA in homeostatic conditions?

We appreciate the reviewer's insightful comment. As the reviewer pointed out, cytokine mRNA expression levels are notably higher in Trm cells from the human lung and murine kidney compared to those from the human kidney and colon. This variance could be attributed to several factors. The scRNA-seq data are derived from cells of human kidney, colon, and lung isolated with different methods and mRNA was processed and sequenced under different conditions (the colon and lung data were generated by other groups and are publicly available). Given the inherently low abundance of cytokine mRNA, the library preparation methods and sequencing depth are crucial factors that likely affect the detection sensitivity of these transcripts. Therefore, we urge caution when directly comparing cytokine mRNA expression across different tissues and datasets. Consequently, in this manuscript, we have refrained from speculating on which organs might harbor more functionally poised Trm cells. With this approach we avoid misinterpreting differences in data that are potentially due to disparate sample-processing methods.

Different subsets of skin Trms preferentially secrete type 1 and type 3 cytokines (S Naik et al Nature 2015, Cheuk, et al., 2017), potentially governed by epigenetic wiring (SL Park, et al., 2023). In s Fig 8b, ISR genes expression differs in TNF+ Trm and IL17A+ Trm in human T cells from diseased tissues. Thus, it would be interesting to compare the ISR score between Trm cell subsets and TNF+ Trm and IL17A+ Trm in Fig 5.

We are very grateful to the reviewer for this discerning suggestion. We compared ISR scores between TNF+ Trm cells and IL17A+ Trm cells under healthy and Crohn's disease conditions. As shown below, IL17A+ Trm cells exhibited lower ISR scores than TNF+ Trm cells under healthy conditions, which aligns with the homeostatic role of IL-17A in barrier organs. However, under inflammatory conditions, many TNF+ Trm cells show ISR scores as low as IL17A+ Trm cells, indicating that both TNF+ Trm cells and IL17A+ Trm cells are activated for efficient cytokine production. (Please see Figure below, which is only shown for reviewers.)

Comparison between colonic TNF+ Trm cells and IL17A+ Trm cells in healthy and Crohn's disease conditions. Violin plots showing the ISR score in TNF+ Trm cells and IL17A+ Trm cells under healthy and Crohn's disease conditions.

The authors found that CD8 Trm cells were poised to type 1 interferons and based the manuscript on the more multipotent CD4 Trm cell population in kidney. Regardless, it would be interesting to know if the same mechanisms is regulating cytokine translation in CD8 T cells, potentially by in vitro experiments mirroring the data in Figure 4 a-f as well as providing analyses of the CD8 compartment in figure 5.

We sincerely appreciate the valuable comments and suggestions. As pointed out by the reviewer, *IFNG* mRNA levels were higher in CD8⁺ T cells than in CD4⁺ T cells. In response to the reviewer's comment, we performed an additional analysis of CD8⁺ Trm cells in our scRNA-seq data. Similar to the analysis of CD4⁺ T cells, CD8⁺ Trm cells in ANCA-GN showed significantly lower integrated stress response scores, indicating CD8⁺ Trm cell activation (as shown in the figures below, only for the reviewers).

Moreover, we isolated CD8⁺ Trm cells from the small intestine of healthy mice and confirmed that the activation of T cells led to eIF2 α dephosphorylation. As observed in CD4⁺ Trm cells, cytokine production from CD8⁺ Trm cells was significantly suppressed by the ISR inducer sodium arsenite, indicating that the ISR pathway is also essential in the functional regulation of CD8⁺ Trm cells.

Given that CD4⁺ T cells are vital and dominant cell types in our renal Trm-cell induction model by the infection of *S. aureus* and in autoimmune diseases analyzed in the manuscript (ANCA-GN and Crohn's disease), a further analysis of CD8⁺ T cells (although of great interest) is beyond the scope of the current study. This point is discussed on page xxx in the revised manuscript, with the data included in revised Extended Data Fig. 10a-c.

CD8⁺ Trm cells are also regulated by the ISR pathway. (A) UMAP plot showing different CD8⁺ T cell clusters. (B) Violin plot showing the ISR score in CD8⁺ Trm cells under healthy and ANCA-GN conditions. (C) Bar graphs showing IFN- γ production and phospho-eIF2 α levels in CD8⁺ Trm cells following stimulation in the presence of the ISR inducer NaAsO₂.

Could the authors expand on the therapeutic strategy in figure 4i-n? From the figure legend, the glomerulonephritis is induced after 2 months (60 days) and treatment given day 70-80, ie 10 days after onset of disease. Is this a model of chronic glomerulonephritis? Would it be possible to revert pathology by initiating treatment in a chronic inflammatory stage of the model? Following cessation of therapy, would the inflammation and pathology relapse?

We appreciate this reviewer's questions. The misleading expression in the figure legend was corrected. Although we stated "after 2 months" in the figure legend, we started the glomerulonephritis model at day 70. Our model is an acute disease model, and intervention started from the onset of the nephritis. We clarified the time course of this experiment in the revised manuscript. As suggested by the reviewer, investigating a chronic disease model is potentially interesting. However, no well-established chronic glomerulonephritis models are available.

Furthermore, in chronic disease models, Trm cells are likely replaced over time by effector T cells migrating to the kidney. Thus the model becomes more complex and it will be technically challenging to investigate the role of previously established renal Trm cells.

Minor comments

Inconsistent CCR7 CITE-seq protein and gene expression in Fig1E heatmaps. Authors may consider checking the positive calling of CCR7+ cells compared to background based on raw CITE-seq protein counts.

We checked the raw CITE-seq protein counts again. The result, however, remained the same. We do not have a clear answer for the discrepancy in CCR7 expression between protein and RNA (the gene and protein data of CD62L and CCR7 are shown below). In this study, we annotated clusters based on the expression levels of *SELL*, CD62L (CITE-seq), and *CCR7*.

Tcm cells-associated gene and protein levels. UMAP plots show the mRNA and protein levels of *SELL*/*CD62L* and *CCR7* corresponding to revised Figure 1e.

Clarify type of data shown in the figure legend of Fig 3B.

We apologize for the lack of clarity. Fig 3B is derived from scRNA-seq analysis of human T cells. We included this information in the revised figure legend.

Same colour code to be used for Tcm, Tem and Trm in Fig 2 and Supplementary Fig 4 for better clarification.

For better clarity, we used the same color code in revised Fig. 2 and Extended Data Fig. 4.

CITE-seq protein name of CD62L should be used instead of gene name, *SELL* in Fig 1, Supplementary Fig S1 and Fig 5.

In revised Fig 1, Extended Data Fig 1, and Fig 5, we replaced the gene name (*SELL*) with the protein name (CD62L).

CITE-seq proteins, CD44 and CCR7 expressions are not shown in supplementary Fig 1C.

The revised Extended Data Fig 1C includes the CITE-seq protein data of CD44 and CCR7.

IL17A, IL17F, IFNG or TNF gene expressions are not shown in Fig 5b.

IL17A, IL17F, IFNG, and TNF gene expressions are now included in revised Fig 5b.

Type 1 and 3 cytokine responsive genes score shown for ANCA-GN in Fig 5F is not shown in the context of Crohn's disease

We have analyzed a new bulk RNA-seq dataset of colon tissues from Crohn's disease and healthy individuals, and calculated the type 1 and 3 cytokine-responsive gene scores for Crohn's disease and showed that the scores are significantly higher in the disease group compared to the control group. This finding is in line with the lower ISR score in Crohn's disease, which would lead to increased cytokine production. We have included the new data in revised Fig 5k.

Type 1 and 3 cytokine-responsive genes score
- Colon tissue

Type 1 and 3 cytokine-responsive gene scores in the colon are elevated in Crohn's disease. Violin plots showing the type 1 and 3 cytokine-responsive gene scores calculated based on colon bulk RNA-seq data (55 samples in the control group and 92 samples in Crohn's disease patients).

Comments from the Editor:

Thank you for your response to the referees' comments on your Article "The integrated stress response/eIF2 α pathway controls cytokine production in tissue-resident memory CD4⁺ T cells" here at Nature Immunology. Although we are interested in the possibility of publishing your study in Nature Immunology, the issues raised by the referees need to be addressed.

Please revise along the lines specified in your letter. Please include Trm cells from the kidney in all the required experiments, as indicated, and please perform at least three replicates for all experiments, including the detection of p-eIF2 α . In addition, we consider it is essential to increase the novelty of the manuscript beyond previous observations that this type of regulation of the mRNA pool has been described in memory T cells. This could be achieved by assessing the regulatory process described here in all memory T cell subsets and in Trm cell across multiple tissues, by including the characterization of multiple cytokines and chemokines and by addressing experimentally the issues raised by referee #1 regarding the specificity of this process compared to translation of other mRNAs, redundancy with other translational control mechanisms and the effect of the size of the untranslated cytokine mRNA pool. As suggested, please revise to accurately describe your results and their implications in the manuscript text. At resubmission, please include a "Response to referees" detailing, point-by-point, how you addressed each referee comment. If no action was taken to address a point, you must provide a compelling argument. This response will be sent back to the referees along with the revised manuscript. We look forward to seeing the revised manuscript and thank you for the opportunity to review your work.

We thank the editor for the helpful comments and advice. In the revised manuscript, we included data for Trm cells from the kidney for all experiments, repeating experiments three times. Moreover, we described the methods of our experiments in more detail, including the tissue source of the T cells, mouse lines, and T-cell stimulation conditions (please see new figure legends).

The revised manuscript clarifies that our study shows, for the first time, that Trm cells represent the main T cell subset that stores cytokine and chemokine mRNA under homeostatic conditions, and that the translation of cytokine mRNA is repressed by the integrated stress signaling pathway (ISR). In line with these results, our experiments stimulating memory T cells in the presence of actinomycin D demonstrate that Trm cells, but not effector or other memory T cells, can rapidly produce cytokine proteins by translating the preformed cytokine mRNAs. These findings underscore that cytokine mRNA storage for immediate cytokine production is a unique finding in Trm cells. As such, the novelty of our work is that Trm cells across different organs store abundant cytokine transcripts with concurrent translation suppression by the ISR, priming them for rapid cytokine protein production and effective effector function upon activation. We further emphasized this point in the discussion of the revised manuscript (please see page 16).

Moreover, we discussed in more detail the potential limitations regarding specificity (Trm vs non-Trm) and potential redundancy with respect to other translational control mechanisms. In addition, we modified our conclusions in the abstract, introduction, and discussion in line with the results.

Reviewer #1

(Remarks to the Author)

I appreciate the authors experiments with ISRIB which appear to show that inhibiting the ISR promotes IFN γ and IL-17A protein production. I agree with the authors that this appears to be a “relatively weak effect”. For clarity of interpretation, I would expect to see a representative FACS plot as the 1% population could either be a very small change in MFI e.g. a “shoulder of a histogram” or a clearly defined population and seeing this essential for interpretation. Furthermore, with such a small change I would expect to see a positive control (e.g., the same cells activated with PMA/ionomycin) to clearly show what range is expected. This is even more important as within this manuscript as TRMs are not always isolated from the same tissue and there are clear differences in response between different experiments.

We appreciate the reviewer's suggestion. To address these concerns, we provide representative FACS plots for the ISRIB-stimulated group and the PMA/ionomycin-stimulated group as a positive control (as shown below, new sFig6i). These FACS plots now provide a more straightforward interpretation of the cytokine-producing populations.

However, while the PMA/ionomycin control will demonstrate the full activation potential of the cells, we emphasize that it cannot be used to estimate the effect size of ISRIB (which was asked for in the following comment). The reason is that PMA/ionomycin induces both mRNA transcription and translation, whereas ISRIB specifically affects translation control.

New sFig6i: Cytokine production in kidney Trm cells treated with ISRIB or PMA/ionomycin

Kidney CD4⁺ Trm cells were left without stimulation or were stimulated with ISRIB overnight or PMA/ionomycin for 2 hours and stained for IL-17A or IFN- γ .

Overall, I am wondering about the differences in some experiments with regards to the frequency of cytokine production in Trms. This makes evaluation of the effect size of some of the pharmacological experiments challenging. It would be helpful if the ISRIB experiments would be presented in a way that one could relate the effect size of the usage of ISRIB on cytokine production to for example Trm activation. What is the proportional effect of ISR on translational suppression? If it is only minor as the authors suggest the claims in the introduction and abstract need to be readjusted and discussed critically.

In our study, we only compared cytokine frequencies between different groups within the same experiment using the same experimental conditions, resulting in consistent findings. Yet,

we agree that the extent of cytokine production after T cell stimulation varies between independent experiments. The experiments are highly standardized in our lab to achieve highly reproducible results. However, some factors cannot be fully controlled, such as the quality of reagents (FCS, nephritic serum, etc.). This was particularly true when Trm cells were isolated from other tissues and mouse strains. Therefore, the experiments always differ to some extent regarding the total amount of cytokine production or the frequency of cytokine-producing Trm cells. However, please note that we did not directly compare the cytokine production of Trm cells between different tissues and mouse lines. Therefore, the differences between the different tissues and mouse lines have no direct influence on our study's results and biological findings.

While we acknowledge that the ISR pathway is probably not the only pathway involved in controlling mRNA translation in Trm cells, our experiments, using the gain-of-function mutant (eIF2 α S51D) and chemical inhibitors, clearly demonstrate the significant impact of ISR on cytokine production (Fig4e-k). We think the ISR pathway plays a substantial role in regulating cytokine mRNA translation. Although ISRIB alone may not produce an impressive effect when other regulatory mechanisms are active, it is still significant. This point is discussed in the revised manuscript (please see page 18).

Why were the experiments done with Trm from small intestine and not from kidney as the other inhibitor experiments?

The reason for starting the inhibitor experiments with intestinal Trm cells was the easier and faster accessibility to this organ compared to other tissues. However, we fully agree with the reviewer that the experiments with Trm cells from kidneys are also of great interest. Accordingly, we now provide results of chemical compound experiments with Trm cells from the kidney in new Fig4c, sFig6c, j, which are in agreement with the results obtained using Trm cells from the small intestine.

Given these concerns, I suggest the manuscript text and abstract be revised to more clearly reflect the data and the limits of its interpretation. Statements in the text especially in the abstract which states "In steady-state conditions, cytokine mRNA translation in Trm cells is suppressed by the integrated stress response (ISR)/ eukaryotic initiation factor 2 α (eIF2 α) pathway" and "These findings suggest that the ISR/eIF2 α pathway tightly controls effector cytokine production in Trm cells, identifying a hitherto unknown mechanism of Trm-cell regulation under homeostatic and inflammatory conditions."

Following the reviewer's suggestion, we modified the abstract and main text (the last part of the introduction, page 2-4) to ensure that the statements more clearly reflect the observed data. Moreover, we now discuss the study's limitations in greater detail, particularly the potential of other pathways to contribute to cytokine regulation in Trm cells. These explanations provide a more balanced interpretation of our findings.

In their response to the review, they write, "The relatively weak effect might be due to other translational regulation mechanisms still being active (e.g., mTOR pathway, RNA-binding proteins, and miRNAs)." I suggest this needs to be clarified in the manuscript.

We clarified this point in the discussion part (please see page 18).

The manuscript discussion would benefit from discussion on how this work fits with prior studies in other non TRM memory T cell types. The regulation of translation of CCL5 (<https://pubmed.ncbi.nlm.nih.gov/12433367/> and <https://pubmed.ncbi.nlm.nih.gov/16983723/> and IFN γ (ref 22) has been shown for other memory subsets thus this mode of regulation of cytokine and chemokine transcripts is likely not a TRM specific process.

We thank the reviewer for bringing up this point. We have included chemokine gene expression data (Fig1h, supplementary Fig1d), and now cite both papers on CCL5 and discussed how our findings align with previous studies on translational regulation in non-Trm memory T-cell subsets. As stated in the manuscript, we do not claim that our study is the first to analyze the translational regulation of cytokine mRNA in T cells. However, in former studies, including those highlighted by the reviewer, the storage levels of cytokine mRNA were not compared across different T-cell subsets or assessed in Trm cells. Most importantly, these studies did not analyze the role of the ISR pathway. In contrast we show by qPCR and scRNA-seq that Trm cells are the dominant T cell subset that stores cytokine mRNA under homeostatic conditions. Experiments stimulating memory T cells in the presence of actinomycin D further demonstrate that Trm cells, but not Tem cells, can rapidly produce cytokine protein by translating the pre-formed cytokine mRNA. scRNA-Seq data, analysis of the phosphorylation status of eIF2 α as well as the pharmacologic inhibition of the ISR identify the ISR/eIF2 α pathway as novel regulatory mechanism for cytokine production of Trm cells. Thus, cytokine mRNA storage under the control of the ISR/eIF2 α pathway appears to be a feature, particularly important for Trm cell function. This attribute of Trm cells is observed across different organs of mouse and human and thus, represents a general regulation mechanism of Trm cells. We discussed this point in more detail in the discussion of the revised manuscript (please see page 16).

I note their comment “Because this manuscript emphasizes cytokine mRNA and does not investigate chemokines”. Is there a biological reason for only focusing on cytokine mRNA?

Cytokines such as IFN- γ or IL-17A are among the most important effector mediators of CD4⁺ T cells (including CD4⁺ Trm cells). Therefore, we focused on these cytokines in renal and intestinal autoimmunity.

In response to the helpful reviewer's suggestion, and based on the manuscripts on CCL5 mRNA mentioned in the previous comment, we included data showing that Trm cells also contain high CCL5 and other chemokine mRNA levels (please see new Fig1h and sFig1d). This finding suggests that the control of mRNA translation in Trm cells is likely not limited to cytokines but also includes chemokines. We revised the manuscript accordingly to highlight that both cytokine and chemokine mRNA are stored in Trm cells, further extending the scope of our findings.

For the effects of inhibitors on activation, they have “focused on signaling mechanisms that are independent from transcription and translation,” but why not provide insight into whether some proteins are able to be newly formed?

We attempted to address the reviewer's comment. Yet, we are unsure about the exact information the reviewer expects us to present. Regarding the effect of chemical inhibitors on T cell activation, we analyzed surface CD62L because its enzymatic shedding upon T cell activation is mediated by the protease ADAM17. This process does not require mRNA transcription or translation. We did not measure activation markers dependent on newly synthesized proteins because the inhibitors we used target mRNA translation. Thus, they would have likely

impacted protein production significantly, potentially complicating the interpretation of those results. We validated our findings by measuring intracellular calcium concentrations, confirming that these chemical inhibitors do not impede T cell activation.

The Trm cell polysome experiments in Fig1m provide indirect evidence that non-cytokine mRNA ATF4 is actively translated under ISR activation in contrast to IL-17, IFN- γ , and CSF2 cytokines.

Regarding the differences p-eIF2 α expression between memory populations, I am not convinced that the new graph indicates that Trms have clearly more p-eIF2 α .

We repeated this experiment three times and confirmed that Trm cells exhibit slightly but significantly higher levels of p-eIF2 α than other memory T cell populations. We acknowledge that the difference was smaller than initially anticipated based on our scRNA-seq analysis. Yet, this result aligns with our observations.

p-eIF2 α levels in different kidney T cell subsets

Histograms showing p-eIF2 α levels in kidney CD4⁺ T cells from three independent experiments. A bar graph of quantification from the third experiment is also shown. For control, MFI levels of anti-rabbit-647 without primary Ab in CD4⁺ CD44⁺ Tmem are shown.

They didn't respond to the original question about cytokine mRNA "It is apparent that the abundance of cytokine mRNA differs between these subsets> is that likely to contribute to the difference in cytokine production?".

We apologize for not addressing this comment earlier. We agree with the reviewer that the abundance of untranslated cytokine mRNA likely contributes to the differences in cytokine production across subsets. As demonstrated in T cell activation experiments in the presence of actinomycin D (Fig2i, j), this untranslated mRNA pool is essential for rapid cytokine production by memory T cells and a unique feature of Trm cells. We discussed this key finding in more detail in the revised manuscript (please see page 16).

Further points for clarification:

Please clarify what is the "control" in figure S6d? Which cell type?

In Figure S6d, the control group represents the 'secondary antibody only' levels. To simplify the visualization, we combined the analysis of different T cell subsets in this control group. Displaying separate negative controls for each subgroup would have made the histogram difficult to interpret because the negative control levels were almost identical across all subsets (figure for reviewer only). Therefore, we combined them to provide a better and more concise representation (please see new sFig. 6d, legend).

Control signal levels in different T cell subsets

Histogram showing APC/AF647 signal levels in different T cell subsets. Cells were stained only with the secondary antibody (anti-rabbit AF647) in the absence of the primary antibody, serving as a control.

From what I can tell the representative FACS plot for part C (also in extended fig 6c) they have switch to using a linear scale for phospho-Eif2a, given that linear scales are very unusual I would expect this to made clearer.

This point is well taken. We used a linear scale for phospho-eIF2α FACS plots to improve the visualization of the data, allowing for a better separation of populations. Now we have provided new data using Trm cells isolated from the kidney with p-eIF2a levels shown in log scale which is more appropriate for FACS data (sFig6c).

p-eIF2α levels in Trm cells

Histogram and bar graph showing p-eIF2α levels in unstimulated Trm cells and IFN-γ or IL-17A -producing Trm cells following 2h stimulation with PMAionomycin.

In figure 2i,j it is not clear whether the Trms from renal and small intestine tissue from the different mouse models were stimulated with the same conditions since the frequencies of IL17 producing cells differ significantly.

We specified the tissue source of Trm cells in the revised manuscript in each figure legend. Kidney and intestinal Trm cells were stimulated under the same conditions. The difference in the frequency of IL-17A-producing cells between the kidney and the intestine is mainly due to using different mouse models. Specifically, we used IL-17A fate reporter mice (*Il17a*-Cre x *Rosa26^{eYFP}*) for the kidney Trm cells, while wildtype mice were used for the intestinal Trm cells. We explained this distinction in the revised manuscript (please see figure legends for Fig2i,j).

Was the origin of Trm cells on which Sephin and Raphin used the same (Small intestine vs renal tissues)? This is not clearly explained in the figure legends and text. inhibitors were used on cells from small intestine but what about the other experiments in figure 4?

For all experiments in the new Figure 4, including the Sephin and Raphin experiments, we used kidney Trm cells (Fig4a, b, d, f). We clarified the source of Trm cells in all figure legends in the revised manuscript.

Overall, this is an import study, but there remain significant question marks with regards to specificity (TRMs vs non TRMs, cytokine mRNA vs overall translation) and redundancy with other translational control mechanisms. The conclusions drawn in the abstract are too strong based on what is shown, and the authors acknowledge this in the response to the reviews, but I do not find them to be sufficiently considered in the discussion.

We thank the reviewer for highlighting the importance of this study. In the revised manuscript, we discussed the potential limitations regarding specificity (Trm vs. non-Trm) and potential redundancy with respect to other translational control mechanisms in more detail. Furthermore, we modified our conclusions in both the abstract and discussion in accordance with the results.

Reviewer #2

(Remarks to the Author)

The authors have taken all my comments into account in the revised version.

Thank you very much!